

# Potential of TROPOMI for understanding spatio-temporal variations in surface NO$_2$ and their dependencies upon land use over the Iberian Peninsula

Hervé Petetin[1], Marc Guevara[1], Steven Compernolle[2], Dene Bowdalo[1], Pierre-Antoine Bretonnière[1], Santiago Enciso[1], Oriol Jorba[1], Franco Lopez[1], Albert Soret[1], and Carlos Pérez García-Pando[1,3]

[1]Barcelona Supercomputing Center, Barcelona, Spain
[2]Royal Belgian Institute for Space Aeronomy (BIRA-IASB), Ringlaan 3, 1180 Uccle, Belgium
[3]ICREA, Passeig Lluís Companys 23, 08010 Barcelona, Spain

**Correspondence:** Hervé Petetin (herve.petetin@bsc.es)

**Abstract.** In orbit since late 2017, the Tropospheric Monitoring Instrument (TROPOMI) is offering new outstanding opportunities for better understanding the emission and fate of nitrogen dioxide (NO$_2$) pollution in the troposphere. In this study, we provide a comprehensive analysis of the spatio-temporal variability of TROPOMI NO$_2$ tropospheric columns (TrC-NO$_2$) over the Iberian Peninsula during 2018-2021 (considering the TrC-NO$_2$ PAL product recently developed using a single TROPOMI
processor version, thus ensuring consistency over the time period). We complement our analysis with estimates of NOx anthropogenic and natural soil emissions. Closely related to cloud cover, the data availability of TROPOMI observations ranges from 30-45 % during April and November to 70-80 % during summertime, with strong variations between northern and southern Spain. Strongest TrC-NO$_2$ hotspots are located over Madrid and Barcelona, while TrC-NO$_2$ enhancements are also observed along international maritime routes close the strait of Gibraltar, and to a lesser extent along specific major highways. TROPOMI
TrC-NO$_2$ appear reasonably well correlated with collocated surface NO$_2$ mixing ratios, with correlations around 0.7-0.8 depending on the averaging time.

We investigate the changes of weekly and monthly variability of TROPOMI TrC-NO$_2$ depending on the urban cover fraction. Weekly profiles show a reduction of TrC-NO$_2$ during the weekend ranging from -10 to -40 % from least to most urbanized areas, in reasonable agreement with surface NO$_2$. In the largest agglomerations like Madrid or Barcelona, this weekend effect
peaks not in the city center but in specific suburban areas/cities, suggesting a larger relative contribution of commuting to total NOx anthropogenic emissions. The TROPOMI TrC-NO$_2$ monthly variability also strongly varies with the level of urbanisation, with monthly differences relative to annual mean ranging from -40 % in summer to +60 % in winter in the most urbanized areas, and from -10 to +20 % in the least urbanized areas. When focusing on agricultural areas, TROPOMI observations depict an enhancement in June-July that could come from natural soil NO emissions. Some specific analysis in Madrid show that the
relatively sharp NO$_2$ minimum used to occur in August (drop of road transport during holidays) has now evolved into a much broader minimum partly de-coupled from the observed local road traffic counting; this change started in 2018, thus before the COVID-19 outbreak.

All in all, our study illustrates the strong potential of TROPOMI TrC-NO$_2$ observations for complementing the existing surface



NO$_2$ monitoring stations, especially in the poorly covered rural and maritime areas where NOx can play a key role, notably for
the production of tropospheric O$_3$.

## 1 Introduction

Nitrogen dioxide (NO$_2$) is a harmful trace gas emitted from both anthropogenic (incomplete combustion processes) and natural
(soils and lightning) sources that plays a key role in tropospheric chemistry, especially in the formation of ozone (O$_3$) and sec-
ondary aerosols. Given its relatively short chemical lifetime, NO$_2$ is often used to investigate the spatio-temporal variability of
emissions from prominent sectors such as road transport or industry. The monitoring of surface NO$_2$ pollution essentially relies
on official air quality (AQ) surface stations. These reference observations benefit from good precision, high temporal resolu-
tion (typically hourly) and long-term time coverage (exceeding 10 years for many stations across Europe). Nonetheless, they
suffer from persistent limitations, including (1) the sparsity of existing AQ networks (e.g. 1 station per 1056 km$^2$ on average
over the Iberian Peninsula in 2018-2021), (2) the low-to-moderate accuracy induced by the intrinsic systematic uncertainties
of the commonly used chemiluminescence-based measurement technique prone to positive artefacts related to NOz species,
especially in rural areas (Dunlea et al., 2007; Villena et al., 2012, e.g.), and to a lesser extent (3) the potential inconsistencies
among the stations as they are operated by numerous different teams. In this context, satellite observations offer a valuable
complement for monitoring the spatio-temporal variability of NO$_2$ by providing consistent NO$_2$ measurements relying on one
single instrument, with full geographical coverage under cloud-free conditions.

Built upon the heritage of Aura OMI (Levelt et al., 2018), Envisat SCIAMACHY (Bovensmann et al., 1999) and MetOp-
A/B/C GOME-2 (Valks et al., 2011) missions, the Tropospheric Monitoring Instrument (TROPOMI) on-board the Copernicus
Sentinel-5 Precursor (S5P) satellite is a last-generation nadir viewing shortwave spectrometer able to measure with high sensi-
tivity and spatial resolution in the ultraviolet-visible, near infrared and shortwave infrared (Veefkind et al., 2012). Among other
key chemical species, TROPOMI is providing groundbreaking information on NO$_2$ tropospheric column (hereafter referred
to as TrC-NO$_2$) and thus offers unprecedented opportunities for monitoring and investigating NO$_2$ pollution and sources. In
the ultraviolet, TROPOMI provides observations at a better signal-to-noise ratio than its predecessor OMI (van Geffen et al.,
2022b; De Smedt et al., 2021), with an improvement of spatial resolution of about a factor of 16. Among other, it has been used
to map industrial point sources (e.g., Griffin et al., 2019; Beirle et al., 2021), identify soil NO emissions (Huber et al., 2020),
investigate the impact of the COVID-19-related lockdown (e.g., Bauwens et al., 2020; Barré et al., 2021), or analyse lightning
NOx emissions (Pérez-Invernón et al., 2022).

The present study aims at characterizing the spatio-temporal variability of TrC-NO$_2$ as seen by TROPOMI over the period
2018-2021. We focus on the Iberian Peninsula including both Spain and Portugal where more than 80 % of the surface mon-
itoring stations keep reporting NO$_2$ (and O$_3$, on which NOx play a key role) levels well above the guidelines recommended
by the World Health Organization (Bowdalo et al., 2022). Our study provides a detailed analysis of the spatial distribution of
NO$_2$, along with its monthly and weekly variability (so-called weekend effect) at both regional and city scales. Such a com-
prehensive observation-based exploration of the spatio-temporal variability of NO$_2$ pollution represents a first crucial step for





better identifying and characterizing the NOx emission sources of main importance over the peninsula. We show how temporal variability of TROPOMI TrC-NO$_2$ depends upon the urban land cover fraction, as well as the crops cover fraction (specifically for the monthly variability). Our analysis of TROPOMI data is complemented with the in-situ NO$_2$ observations available at

the surface, and estimates of primary NOx anthropogenic and natural soil emissions. Considering the persistent uncertainties affecting emission inventories, a detailed analysis of the weekly variability provides key information about the joint contribution of specific emission sectors (in this case, commuting-related road transport and part of the industries) relative to total emissions. Overall, our study also aims at shedding light on specific patterns of interest, then susceptible to guide diagnostic-oriented chemistry-transport model (CTMs) evaluations. Given the specificities of TROPOMI observations, first and foremost their columnar nature and their incomplete sampling, analysing space-based TrC-NO$_2$ jointly with surface-based NO$_2$ is also

key for assessing the potential and limitations of such data to reliably describe the NO$_2$ variability prevailing at the surface where most physico-chemical processes and adverse impacts of NO$_2$ pollution are of strongest importance.

The dataset and methods are introduced in Sect. 2. Results are presented in Sect. 3. An overall discussion and some conclusions are given in Sect. 4.

## 2 Data and methods

### 2.1 TROPOMI TrC-NO$_2$ data

Launched in late 2017, the TROPOMI instrument on-board the S5P satellite provides daily tropospheric column measurements of several important trace gases including NO$_2$, at an overpass time of 13h30 local solar time, with global coverage every day (under cloud-free conditions) (Veefkind et al., 2012). TROPOMI TrC-NO$_2$ were initially measured at a spatial resolution of

7.2x3.5 km$^2$ at nadir, refined to 5.6x3.5 km$^2$ from 6 August 2019 on-wards. More information on the typical dimensions of TROPOMI pixels along an orbit is given in Table 1. On average, this change of resolution reduced the mean TROPOMI pixel area from 43 to 34 km$^2$ (-22 %).

TrC-NO$_2$ observations are publicly delivered as L2 products along S5P orbits on the S5P hub (https://scihub.copernicus.eu/, https://doi.org/10.5270/S5P-s4ljg54 for v1, https://doi.org/10.5270/S5P-9bnp8q8 for v2; last access : 10/03/2022). Besides

the near-real-time (NRTI) TrC-NO$_2$ products that are not considered in this study, two TROPOMI TrC-NO$_2$ products are available on the S5P hub, including (1) the so-called reprocessed (RPRO) TrC-NO$_2$ products that cover the period 30/04/2018-17/10/2018, and (2) the so-called offline (OFFL) TrC-NO$_2$ products that cover the period 17/10/2018-present, although with different processor versions. More details about the TROPOMI NO$_2$ products can be found in the Algorithm Theoretical Basis Document (van Geffen et al., 2022a), the Product User Manual (Eskes et al., 2022) and the Product Readme File

(Eskes and Eichmann, 2022). In order to overcome the inconsistencies introduced by the continuous change of processor versions, the historical TROPOMI TrC-NO$_2$ dataset has been reprocessed by the end of 2021 using the last processor version available at that time (namely the version 2.03.01). Denominated PAL (Eskes et al., 2021) (and freely available on https://data-portal.s5p-pal.com/; last access : 21/06/2022), this new temporally consistent product thus covers the period 01/05/2018-14/11/2021 and can be combined with OFFL products beyond 14/11/2021 as both are produced with the same pro-



cessor version (until 17/07/2022 when a new version 2.04.00 was introduced). Note that the L1b versions used for PAL+OFFL are not fully consistent, but the impact on TROPOMI TrC-NO$_2$ is very small (Eskes et al., 2021). The processor used to generate PAL products solved some issues notably related to cloud properties, the cloud pressure being overestimated in the oldest data versions (Compernolle et al., 2021; van Geffen et al., 2022b).

In the present study, we used this PAL TrC-NO$_2$ products combined with OFFL products after 14/11/2021 (until 31/12/2021,

end of our period of study). Our TROPOMI TrC-NO$_2$ dataset can thus be considered as fully consistent. For information purpose, a comparison of PAL and OFFL+RPRO TrC-NO$_2$ dataset during their overlapping period is given in Sect. A in the Appendix. As expected from the PAL documentation, TrC-NO$_2$ show a good consistency with Pearson correlation coefficients (PCC) above 0.98 and slightly higher TrC-NO$_2$ values in PAL, especially in most polluted areas and/or during wintertime (as shown by normalized mean biases between +1 and +5 % depending on the season, and linear regression slopes around

100   1.02-1.09).

To facilitate analysis, we gridded the L2 products on a fixed regular grid of 0.025°x0.025° longitude-latitude resolution covering the Iberian Peninsula (longitudes range from 10°W to 5°E, latitudes range from 35°N to 45°N). Following the guidelines provided in the Product User Manual, all TROPOMI individual pixels with quality indicator values (*qa_value*) below or equal to 0.75 have been discarded, which removed pixels with too strong cloud coverage, presence of snow or ice, or affected by

other types of retrievals errors. All orbit files were regridded on the target grid using a conservative method and merged through an area-weighted averaging in order to properly account for the overlap of neighbouring orbits at the edges of the swath. All the analysis and comparisons performed in this study are based on these regridded TROPOMI data.

Over the last years, the TROPOMI TrC-NO$_2$ products have been extensively evaluated against ground-based TrC-NO$_2$ observations, but essentially using the OFFL/RPRO products (since PAL products have been delivered only in December 2021).

Based on ground-based MAX-DOAS TrC-NO$_2$ observations at 19 surface stations, a recent and comprehensive evaluation indicated that TROPOMI TrC-NO$_2$ are affected by a substantial negative bias, ranging between -23 and -37 % in clean or slightly polluted areas, and increased to -51 % over highly polluted areas (Verhoelst et al., 2021); regular validation updates can be found in Lambert et al. (2022). Among other reasons (e.g. differences in representativeness, treatment of clouds and aerosols), a substantial part of this negative bias is attributed to the overly coarse a priori profiles used in the S5P retrieval algorithm and

currently obtained from a 1°x1° global simulation of the massively parallel (MP) version of the Tracer Model version 5 (TM5) (Verhoelst et al., 2021). During the S5P validation campaign over Belgium using airborne observations, Tack et al. (2021) were indeed able to reduce the negative bias of TROPOMI TrC-NO$_2$ over two large Belgium cities from -14 to -1 % using a priori profiles from the Copernicus Atmospheric Monitoring Service (CAMS) regional ensemble (0.1°x0.1° spatial resolution). Consistent with this, the validation presented in Douros et al. (2022) using European ground-based MAX-DOAS and Pandora

instruments demonstrated that the negative bias was improved by 5-18% when using the CAMS a priori profiles. In the present study, we kept the original TROPOMI products, although we acknowledge that near future studies should probably explore such a use of alternative a priori profiles.





**Table 1.** Statistics on the pixel dimensions of a given TROPOMI L2 file before and after 6 August 2019. pX here corresponds to the X[th] percentile.

| Period | Metric | Size pixel along swath (km) | Size pixel along scanline (km) | Area (km$^2$) |
|---|---|---|---|---|
| Before 05/08/2019 | mean | 6.1 | 7.1 | 43.3 |
| | min | 3.6 | 6.7 | 26.0 |
| | p5 | 3.6 | 6.9 | 26.2 |
| | p25 | 3.9 | 7.1 | 28.2 |
| | p50 | 5.0 | 7.2 | 36.0 |
| | p75 | 7.8 | 7.2 | 54.8 |
| | p95 | 11.8 | 7.2 | 82.9 |
| | max | 15.0 | 7.2 | 106.4 |
| After 06/08/2019 | mean | 6.1 | 5.5 | 33.7 |
| | min | 3.6 | 5.2 | 20.2 |
| | p5 | 3.7 | 5.4 | 20.5 |
| | p25 | 3.9 | 5.5 | 22.0 |
| | p50 | 5.0 | 5.6 | 28.0 |
| | p75 | 7.8 | 5.6 | 42.6 |
| | p95 | 11.8 | 5.6 | 64.6 |
| | max | 15.0 | 5.6 | 83.0 |

## 2.2 Surface air quality observations

Hourly observations of surface $NO_2$ and $O_3$ mixing ratios are taken from the European Environmental Agency (EEA) AIR-
BASE and AQ eReporting (EEA, 2020); $O_3$ observations in this study are only briefly used in Sect. 3.1 for discussing the avail-
ability of TROPOMI TrC-$NO_2$ observations during $O_3$ episodes. A quality assurance procedure is applied using the GHOST
(Globally Harmonised Observational Surface Treatment) metadata, GHOST being a project developed at the Earth Sciences
Department of the Barcelona Supercomputing Center that aims at harmonizing global surface atmospheric observations and
metadata, for the purpose of facilitating quality-assured comparisons between observations and models within the atmospheric
chemistry community. More details on the quality assurance filtering are given in Appendix C. Based on these hourly observa-
tions, a minimum data availability criteria of 75 % (i.e. 18 over 24 hours) has been chosen for computing daily-scale statistics.
At hourly (daily) scale, the quality assurance filtering removed 4 % (5 %) of the background stations and 16 % (20 %) of the
$NO_2$ observations. The choice of the most appropriate time scale to consider for the surface-based $NO_2$ in their comparison
against space-based TrC-$NO_2$ is not straightforward given the very different (much larger) spatio-temporal representativeness
of TROPOMI columns compared point stations observations. Therefore, in the present study, we consider several time scales





for surface $NO_2$ mixing ratios, including the daily 24-hour mean $NO_2$ (hereafter referred to as *d* time scale), the daily 1-hour maximum $NO_2$ (*d1max*) and the daily TROPOMI-overpass-time $NO_2$ (*dop*) which corresponds to the hourly $NO_2$ mixing ratio observed around 13h30 local solar time.

To facilitate the analysis and comparisons, the surface $NO_2$ observations are gridded on the same target grid as TROPOMI, with all stations in a given cell averaged together. Given the spatial resolution of TROPOMI TrC-$NO_2$, only background stations are taken into account here, which includes rural, sub-urban and urban background stations, but excludes industrial and traffic stations. This is expected to limit the problem of representativeness when comparing TROPOMI and surface observations.

### 2.3 Anthropogenic and natural soil NOx emissions

Besides surface $NO_2$ and TROPOMI TrC-$NO_2$ observations, we also discuss in this study the variability of NOx emissions. Anthropogenic NOx emissions are taken from the annual European CAMS-REG-AP_v5.1 emission inventory (Kuenen et al., 2022) and preprocessed with the HERMESv3_GR emission model (Guevara et al., 2019), using an updated version (v3.2) of the CAMS-REG-TEMPOv2.1 temporal profiles described in Guevara et al. (2021). Compared to the CAMS-REG-TEMPOv2.1 that was relying on traffic counts data, the update proposed in CAMS-REG-TEMPOv3.2 notably includes an improvement of the road transport emission temporal profiles through the use of TomTom traffic congestion statistics from about 50 countries (https://www.tomtom.com/traffic-index/). For a more detailed analysis over industrial areas, the HERMESv3 Spanish industrial point source database is considered. The inventory reports exact geographic location and hourly emissions per individual facility based on the national reporting of air pollutant emissions from large point sources, the PRTR-Spain database and activity-based temporal profiles used to downscale original annual emissions to the hourly level (Guevara et al., 2020).

Besides anthropogenic NOx emissions, NO is also emitted by soils through complex microbial (e.g., nitrification and denitrification) and chemical processes, notably controlled by nitrogen inputs to the ecosystem, temperature, soil water content and soil pH (Butterbach-Bahl et al., 2013). Here, we computed the natural soil NO emissions in 2019 with the Model of Emissions of Gases and Aerosols from Nature (MEGAN) version 2.1 (Guenther et al., 2012), fed by meteorological input obtained from the WRF-ARW version 3.6 model (Skamarock et al., 2008) configured as described in Pay et al. (2019).

### 2.4 Ancillary data

In this study, $NO_2$ observations are analysed in combination with land cover data, in order to further investigate how the weekly and monthly variability of $NO_2$ varies spatially depending on the local environment. These land cover data are taken from the high-resolution (native resolution of 100-m) global-scale Copernicus Land Monitoring Service land cover dataset (https://land.copernicus.eu/global/products/lc, last access : 20/01/2020), and include information about urban area, bare, crops, grass, moss, shrub, snow, tree and water (Buchhorn et al., 2020). They are currently obtained from the PROBA-V space sensor that should be replaced in the near-future by data from the Sentinel-2 satellite.

To help the interpretation of specific situations, we also sporadically use some meteorological information (e.g. 2-m temperature, 10-m surface wind speed) taken from the ERA5 reanalysis (Hersbach et al., 2020) provided by the European Centre for Medium-Range Weather Forecasts (ECMWF). ERA5 data have a native spatial resolution of about 31 km and 137 vertical



levels, although data were downloaded on a 0.25°x0.25° regular longitude-latitude grid from the Climate Data Store (Coperni-
cus Climate Change Service (C3S), 2017) (https://www.ecmwf.int/en/forecasts/datasets/reanalysis-datasets/era5; last access :
18/07/2022).

To support our discussion, we also use some statistical information about tourism in Spain, including the monthly inter-regional
movements of Spanish residents publicly available in the ETR database and the monthly arrival of international tourists publicly
available in the FRONTUR database (both databases being freely available at https://www.dataestur.es/apidata/; last access :
08/08/2022).

Finally, the maps shown in this study make use of different types of geographical information, including the Nomenclature
of territorial units for statistics (NUTS) administrative boarders shapefiles freely provided by Eurostat (https://ec.europa.eu/
eurostat/web/gisco/geodata/reference-data/administrative-units-statistical-units/nuts; last access date: 11/01/2022), the Span-
ish road network shapefiles obtained from the HERMESv3_BU model database (Guevara et al., 2020) (https://earth.bsc.es/
gitlab/es/hermesv3_bu; last access : 01/08/2022), as well as the GHSL-OECD functional urban areas produced by the European
Joint Research Center (Schiavina et al., 2019) (https://ghsl.jrc.ec.europa.eu/ghs_fua.php; last access : 01/06/2022). Functional
urban areas identify the geographical extent of a given major city and its surrounding administrative units and commuting area,
which typically largely exceeds the limits of the metropolitan area itself.

## 3 Results

We first provide some insights on the impact of cloud cover on the TROPOMI data availability (Sect. 3.1) and analyse the
distribution of TROPOMI TrC-NO$_2$ data over the Iberian Peninsula (Sect. 3.2). We then investigate their correlation with
surface NO$_2$ mixing ratios (Sect. 3.3). We finally characterize the TrC-NO$_2$ variability at both weekly and monthly scales
(Sect. 3.4 and 3.5).

### 3.1 Impact of cloud cover on data availability

Compared to surface monitoring stations, space sensors like TROPOMI benefit from an incomparable geographical coverage
but are unfortunately not able to deliver reliable data in the presence of clouds or snow and ice at the surface. In this section,
we analyse to which extent this limitation impacts the availability of TROPOMI TrC-NO$_2$ data along the year over the Iberian
Peninsula. The monthly-scale availability of TROPOMI TrC-NO$_2$ data over the Iberian Peninsula is reported in Table 2.
Overall, TROPOMI TrC-NO$_2$ observations are available during around 55-60 % of the days over the Iberian Peninsula, with
a relatively similar availability in Spain and Portugal. Data availability depends upon cloud cover and strongly varies along
the year, with lowest values in April and November (~30-45 %), low-to-moderate values in wintertime (~45-60 %) and higher
values in summertime (~70-80 %). The mean annual map shown in Fig. 1 highlights strong regional contrasts, with availability
above 70 % over both the southern part of the peninsula and the arid Ebro Valley within the Aragon region in the northeast, and
around 35 % over the Cantabric coast and the Pyrenees (the availability by region is given in Table E1 in the Appendix). Over
our domain of study, the availability of TROPOMI TrC-NO$_2$ data is highest over the sea/ocean along the Spanish southern





**Table 2.** Mean data availability of TROPOMI TrC-NO$_2$ over Spain and Portugal, on average over 2018-2021.

| Period | Spain | Portugal |
|---|---|---|
| All | 56 % | 60 % |
| January | 46 % | 50 % |
| February | 59 % | 59 % |
| March | 54 % | 56 % |
| April | 29 % | 42 % |
| May | 52 % | 59 % |
| June | 60 % | 65 % |
| July | 77 % | 76 % |
| August | 77 % | 77 % |
| September | 61 % | 66 % |
| October | 58 % | 60 % |
| November | 42 % | 47 % |
| December | 47 % | 50 % |

coast.

Despite some substantial data gaps due to cloud cover, TROPOMI TrC-NO$_2$ data benefit from a high availability during O$_3$ episodes in the Iberian Peninsula, which typically occur under anticyclonic conditions with low cloud cover. Indeed, the TROPOMI TrC-NO$_2$ availability at days and locations where the so-called O$_3$ target threshold is exceeded (daily 8-h maximum O$_3$ above 120 $\mu$g m$^{-3}$; 14,649 events) is 85 %, and reaches 100 % for the O$_3$ information threshold exceedances (daily 1-h maximum O$_3$ above 180 $\mu$g m$^{-3}$; 22 events). In contrast, the availability of TROPOMI TrC-NO$_2$ data is reduced to 66 % when and where NO$_2$ target thresholds (daily 1-h maximum NO$_2$ above 40 $\mu$g m$^{-3}$; 45,750 events) are exceeded at the surface, essentially because high NO$_2$ episodes occur preferably during the cold season when cloud coverage is typically higher.

### 3.2 Spatial distribution of TROPOMI TrC-NO$_2$

The mean TrC-NO$_2$ spatial distribution over the Iberian Peninsula is shown in Fig. 2. Strong hot-spots appear in the most important cities, first and foremost Madrid and Barcelona, with secondary hot-spots over smaller cities (e.g., Lisbon, Porto, Valencia, Seville, Granada, Bilbao, Gijón). A zoom of the mean TrC-NO$_2$ over the functional urban area of most important Iberian Peninsula cities is given in Fig. 3. Note that these functional urban areas are defined by the European Union as "sets of contiguous local (administrative) units composed of a city and its surrounding, less densely populated local units that are part of the city's labour market (commuting zone)" and thus typically extend much beyond the administrative limits of the agglomeration. On average, maximum TrC-NO$_2$ values reach about 8.5 Pmolec cm$^{-2}$ in Madrid, 7 Pmolec cm$^{-2}$ in Barcelona, 4-4.5 Pmolec cm$^{-2}$ in Lisbon, Valencia, Porto, Granada and Seville, and 2 Pmolec cm$^{-2}$ in Zaragoza and Palma de Mallorca. The



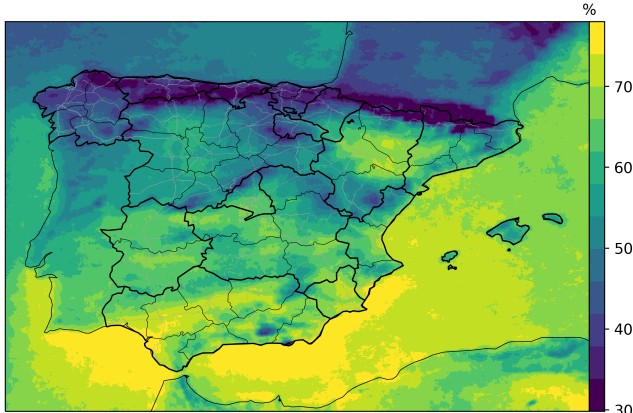

**Figure 1.** Mean data availability of TROPOMI TrC-NO$_2$, on average over 2018-2021. Black and grey lines correspond to administrative borders and Spanish major roads, respectively (sources: see Sect. 2.4).

spatial resolution of TROPOMI is fine enough to reveal the presence of large natural areas within the largest metropolitan areas, including the Casa de Campo park in western Madrid and the Serra de Collserola and Serralada natural areas in Barcelona.

Although not as clearly as in Algeria, a few highways can be distinguished over the Iberian Peninsula, including those linking Zaragoza / Logroño / Vitoria-Gasteiz (AP-68), Valladolid / Vitoria-Gasteiz (A62), and Valencia / Alicante / Murcia (A35-A31-A7-A7S). Regionally, the lowest TrC-NO$_2$ values are found in intermediate- (southern Aragon, eastern Castilla-La-Mancha and eastern Andalucia) and high-altitude regions (Pyrenees, Picos de Europa, Sierra de Gredos at west of Madrid) with low population densities. The major shipping route passing through Gibraltar can be clearly identified, especially in the Mediter-

ranean.

The distribution of TrC-NO$_2$ values at daily, monthly and annual time scales is described in Table 3. The historical maximum TrC-NO$_2$ observed by TROPOMI over the Iberian Peninsula reaches the extreme value of 63 Pmolec cm$^{-2}$, and occurred in early January 2021 in Madrid a few days after Filomena - the largest snowstorm since 1971 - hit central Spain (Tapiador et al., 2021) (see Sect. D in the Appendix for a more detailed discussion of this episode). Apart from this extraordinary situation,

more than 99.9% of the TrC-NO$_2$ values over the Iberian Peninsula typically remain below 11 Pmolec cm$^{-2}$ at daily scale (and below 8 and 7 Pmolec cm$^{-2}$ at monthly and annual scale, respectively). Note that due to noise in the TROPOMI measurements, a few TrC-NO$_2$ values are negative (1.3 % of the daily TrC-NO$_2$ and 0.3 % of the monthly TrC-NO$_2$).

We saw in Sect. 3.1 that missing days in the TROPOMI dataset are not randomly distributed, which may thus bias the climatological averages. In order to provide some insights on this potential issue, we compared the climatological daily mean surface

NO$_2$ considering all available days to the climatological daily mean surface NO$_2$ considering only the days with available TROPOMI observations. The comparison is thus here based on all the cells containing surface monitoring stations (N=283). Both climatologies appear very consistent (PCC=0.99, nRMSE=8 %) but a small positive bias (+4 %) is found to be induced by the TROPOMI missing days. Although we typically expect a negative bias of space-based TrC-NO$_2$ due to the typically larger





**Table 3.** Distribution of TrC-NO$_2$ values (in Pmolec cm$^{-2}$) over the Iberian Peninsula (Spain and Portugal) at daily, monthly and annual scales.

| Metric | Daily | Monthly | Annual |
|---|---|---|---|
| mean | 1.4 | 1.4 | 1.4 |
| std | 1.0 | 0.7 | 0.5 |
| p0 | -4.8 | -2.2 | 0.2 |
| p1 | -0.1 | 0.6 | 0.8 |
| p5 | 0.3 | 0.8 | 0.9 |
| p25 | 0.9 | 1.1 | 1.1 |
| p50 | 1.3 | 1.3 | 1.3 |
| p75 | 1.7 | 1.6 | 1.5 |
| p95 | 2.9 | 2.5 | 2.2 |
| p99 | 4.9 | 4.0 | 3.5 |
| p99.5 | 6.3 | 4.9 | 4.3 |
| p99.9 | 11.1 | 8.0 | 6.5 |
| p100.0 | 62.8 | 21.9 | 10.2 |
| N | 76,923,523 | 4,405,167 | 301,107 |

cloud cover during wintertime when NO$_2$ levels are highest (as found in e.g., Compernolle et al. (2020) for OMI), our results
with TROPOMI show a small overestimation, which seems to be due to the specificity of the TROPOMI data availability over
the Iberian Peninsula where the minimum availability is reached in April and November rather than during the coldest winter
months.

### 3.3 TROPOMI TrC-NO$_2$ versus surface NO$_2$ mixing ratios

Given its relative short chemical lifetime, NO$_2$ levels remain high close to emission sources, which typically induces a rea-
sonable co-variability of surface NO$_2$ mixing ratios and space-based TrC-NO$_2$. Nonetheless, this co-variability is adversely
affected by the intrinsically different nature and representativeness of both types of measurement, as well as the noise of
TrC-NO$_2$ observations. In this section, we investigate these aspects by analyzing the correlation between TROPOMI-based
TrC-NO$_2$ observations and surface NO$_2$ mixing ratios measured by monitoring stations. Density scatter plots of TrC-NO$_2$
versus surface NO$_2$ at TROPOMI overpass time are given in Fig. 4. In addition, Fig. 5 shows the evolution of the PCC when
averaging TrC-NO$_2$ and surface NO$_2$ over different windows, with a data availability criteria of 50 %. For information purpose,
mean daily time series of TrC-NO$_2$ and surface NO$_2$ mixing ratios averaged over the entire domain are shown in Fig. E5 in the
Appendix. On a daily basis (i.e. window of 1 d), both TrC-NO$_2$ and surface NO$_2$ are reasonably well correlated, with a PCC
of 0.70. When increasing the time window, the PCC progressively increases, which illustrates the progressive improvement of



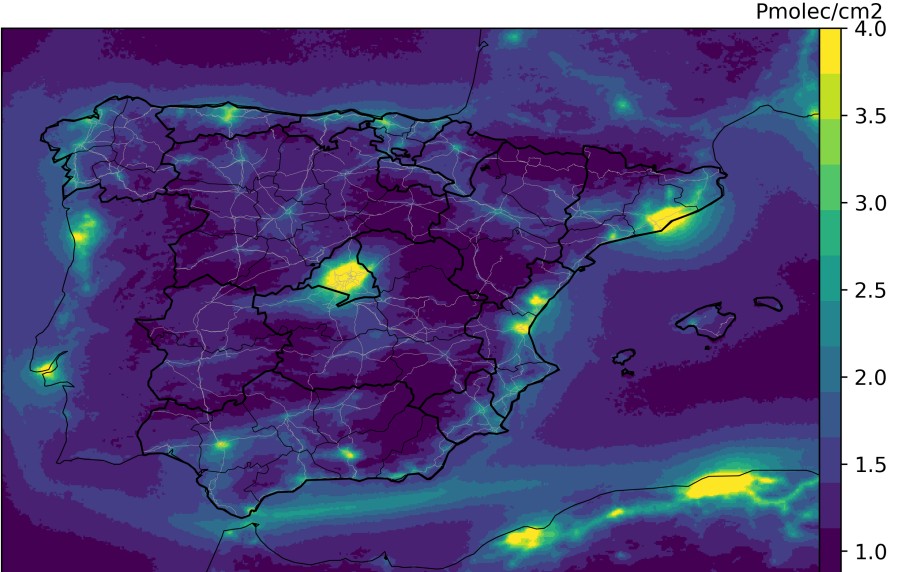

**Figure 2.** Climatological mean of TROPOMI TrC-NO$_2$. Black lines and grey lines correspond to administrative borders and Spanish major roads, respectively (sources: see Sect. 2.4).

the TROPOMI signal-to-noise ratio. It reaches its maximum value at windows of 90-120 d (i.e. 3-6 months) and then starts to decrease, likely due to a too limited number of points. Similarly, the intercept of the linear regression of TROPOMI TrC-NO$_2$ versus surface NO$_2$ progressively improves when averaging over larger windows. Relatively consistent results are obtained for the daily mean and daily 1-h maximum, although PCC values are typically lower especially for the last one.

Other recent studies also investigated the correlations of TROPOMI TrC-NO$_2$ against ground-based TrC-NO$_2$ observations and/or surface NO$_2$ mixing ratios. Ialongo et al. (2020) obtained PCC of 0.68 (N=94) between TROPOMI and PANDORA
ground-based TrC-NO$_2$ in Helsinki during April-September 2018, while the correlation between PANDORA NO$_2$ tropospheric columns and surface NO$_2$ concentrations was also relatively high (PCC=0.74). Jeong and Hong (2021) found a PCC of 0.67 (N=78,048) between TROPOMI TrC-NO$_2$ and surface NO$_2$ concentrations over the 573 monitoring stations available in South Korea, improved to 0.69 (N=70,439) when removing traffic stations. Enhanced correlations were obtained when considering the annual average, with 0.84 (N=573) and 0.88 (N=532) with all and non-traffic stations, respectively. Over northern (10 sta-
tions) and southern Italy (37 stations), Cersosimo et al. (2020) explored the relationship between surface NO$_2$ and TROPOMI TrC-NO$_2$ with and without a kriging-based regridding and spatial interpolation (1x1 km$^2$). They obtained a PCC of 0.71 and 0.65 over northern and southern Italy, respectively, increased to 0.92 and 0.84 with the monthly scale krigged TROPOMI data. These different results are typically in line with the PCC found here over the Iberian Peninsula; note that our results are based on a much larger number of points.

Therefore, although it only measures NO$_2$ tropospheric columns, TROPOMI is able to provide a very useful information



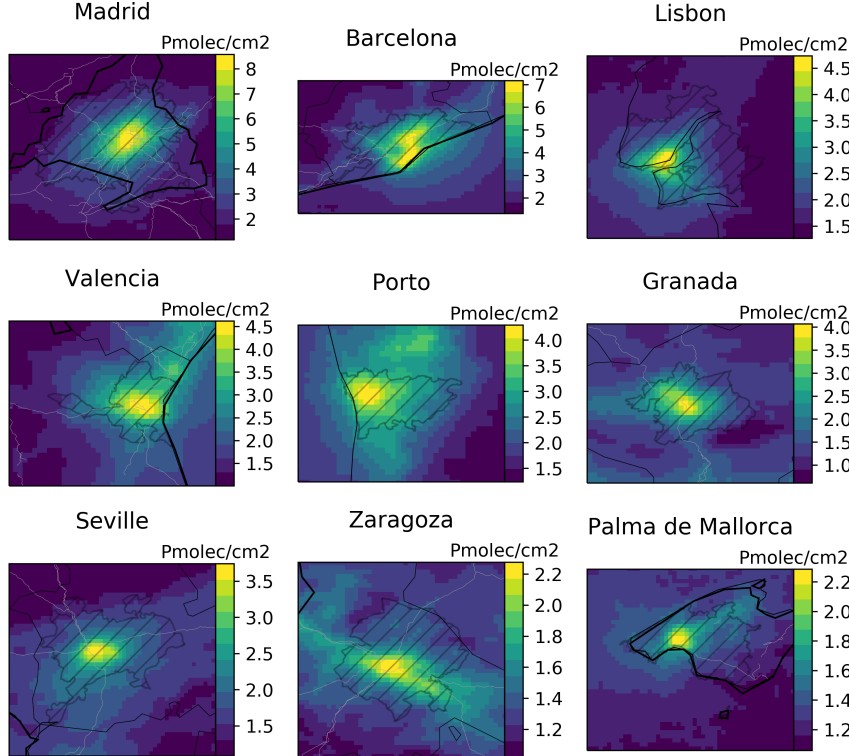

**Figure 3.** Mean TROPOMI TrC-NO$_2$ over main Iberian Peninsula cities. Black lines, grey lines and hatched areas correspond to administrative borders, Spanish major roads and functional urban areas, respectively (sources: see Sect. 2.4). Pixels are shown at their 0.025°x0.025° resolution.

regarding the spatio-temporal variability of surface NO$_2$, especially when considering sufficient large time windows. Nevertheless, the persistent scatter clearly highlights that the relationship between TROPOMI TrC-NO$_2$ and surface NO$_2$ mixing ratios is more complex than a simple linear relationship. Based on the linear regression of daily TrC-N0$_2$ versus surface NO$_2$, we analysed the corresponding residuals (here defined as the distance in Pmolec cm$^{-2}$ between a given point and the linear regression line). Residuals range between -19 and +44 Pmolec cm$^{-2}$, with 1[th], 5[th], 50[th], 95[th] and 99[th] percentiles of -5, -3, 0, +3 and +9 Pmolec cm$^{-2}$, respectively. The high positive residuals correspond to situations where strong TrC-NO$_2$ were measured by TROPOMI, while relatively low NO$_2$ mixing ratios were measured by surface background stations. The location of the 10 % largest positive residuals is shown in Fig. E4 in the Appendix: more frequent during cold months, these residuals occur preferentially over large urban areas (e.g. Madrid, Barcelona). Although they could be explained by the possible presence of NO$_2$ pollution aloft in the boundary layer, they are more probably associated with conditions of low dispersion, leading to substantial spatial heterogeneities of surface NO$_2$ over the TROPOMI pixel area.





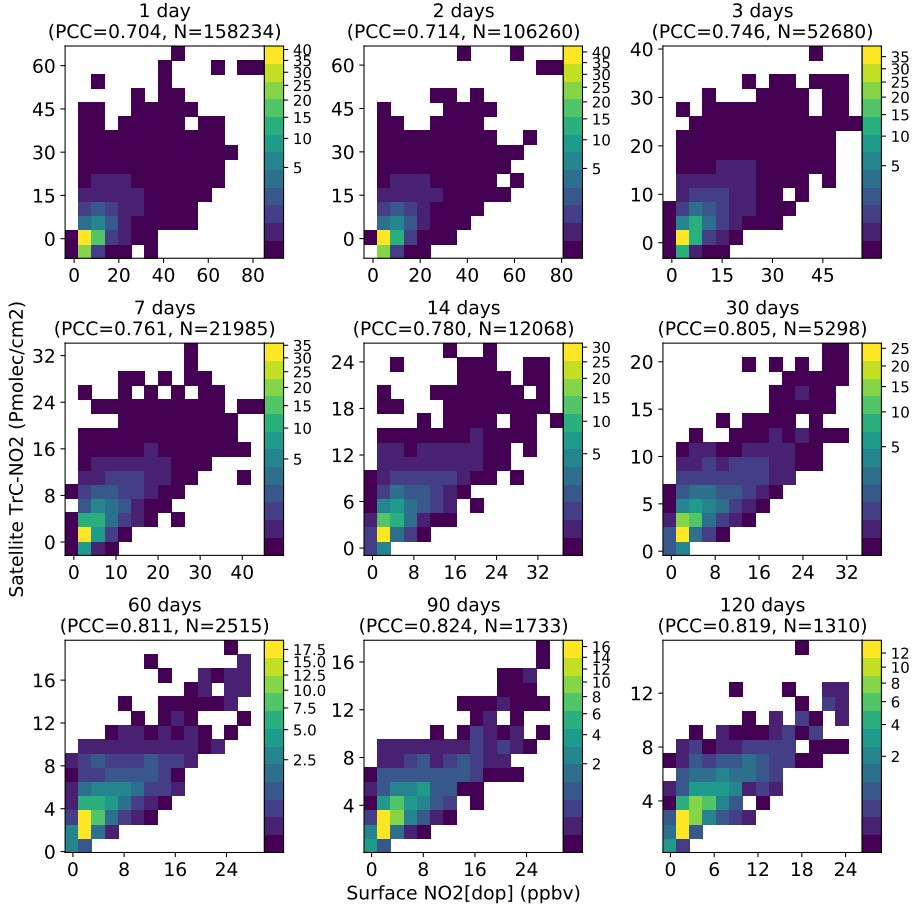

**Figure 4.** Density scatter plot of TROPOMI TrC-NO$_2$ against surface NO$_2$ mixing ratios at TROPOMI overpass time (*dop* time scale), averaged over different time windows.

## 3.4 Weekly variability

### 3.4.1 TrC-NO$_2$ weekly variability and its dependency upon urban land cover fraction

The mean weekly profiles of TROPOMI TrC-NO$_2$ over the Iberian Peninsula are shown in Fig. 6, averaged over cells of
different urban cover fractions. More urbanized areas typically show both higher TrC-NO$_2$ values and stronger reduction during the weekend, relative to weekdays. Hereafter, the mean relative difference of NO$_2$ levels between weekdays and weekend (or only Saturday or Sunday) is referred to as the weekend effect and is calculated as $(\overline{X} - \overline{WD})/\overline{WD} \times 100\%$ with $\overline{WD}$ the mean NO$_2$ during weekdays (Monday to Friday if not specified) and $\overline{X}$ the mean NO$_2$ during the weekend (Saturday-Sunday if not specified) or Saturday or Sunday taken individually. Over the areas exceeding 90 % of urban fraction, this weekend effect



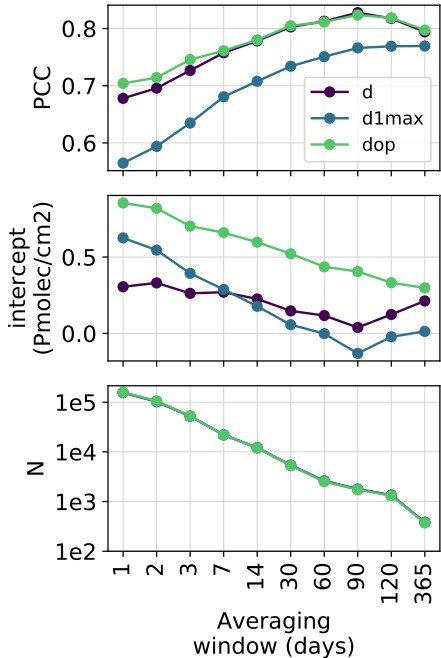

**Figure 5.** Pearson correlation coefficient (top panel), intercept (middle panel) and number of points (lower panel) of the least square linear regression of TROPOMI TrC-NO$_2$ versus surface NO$_2$ mixing ratio for different time windows.

reaches around -30 % during the weekend, with stronger reductions on Sunday (-36 %) than on Saturday (-25 %). Interestingly, it is worth mentioning that the same TROPOMI-based weekly profiles over the first strict COVID-19 lockdown (here defined as the period 15/03/2020-31/05/2020) highlight a few noticeable differences (see Fig. E1 in the Appendix), including (1) a quite substantial NO$_2$ reduction on Friday over the most urbanized area (up to -15 %), (2) a similar NO$_2$ reduction on both Saturday and Sunday whatever the urban fraction (corresponding to a stronger decrease on Saturday compared to the previously

discussed weekly profiles), and (3) a stronger weekend effect over areas below 10 % urban fraction; the same weekly profiles obtained considering only the cells with surface observations available are apparently much less robust likely due to an overly low number of points, but in agreement with the corresponding profiles observed at the surface.

    Over the areas with less than 10 % urban cover, the weekly profile is flattened with weekend reductions around -10 % and only small differences between Saturday and Sunday. Such a persistence of the weekend effect in rural areas can be at least partly

attributed to (1) the transport of NO$_2$ pollution from urban to rural areas leading to a smoother but persistent weekly variability in downwind rural areas, and (2) the weekly variability of total NOx emissions in rural areas. Regarding the variability of emissions, the estimated total (anthropogenic and natural soil) NOx emissions indeed highlight lower emissions during the weekend, around -20 % (Fig. 7). While natural soil NOx emissions do not follow any clear weekly variability, the anthropogenic emissions prevailing in rural areas (from e.g. road and non-road transport, agriculture, isolated industrial facilities) do show a





substantial relative change during the weekend, down to -50 % and are high enough to induce some weekly variability to total NOx emissions, although it is worth reminding here that both anthropogenic and natural soil NOx emissions remain affected by substantial uncertainties. Although not directly comparable, the weekend effect of total NOx emissions is roughly consistent but slightly stronger than the one observed in TROPOMI TrC-NO$_2$, whatever the urban cover fraction.

Among weekdays, a minor but still noticeable variability is observed in TrC-NO$_2$ data, with slightly lower TrC-NO$_2$ values 310    on Monday, and slightly higher TrC-NO$_2$ on Tuesday. NOx emissions also show slightly lower emissions on Monday but also higher ones on Friday, maybe driven by higher traffic intensity and traffic congestion before the weekend. Here, the TROPOMI-based weekly profile might hide this feature due to a too early overpass time (13h30 local solar time).

### 3.4.2   Spatial distribution of the TrC-NO$_2$ weekend effect

In terms of spatial distribution (Fig. 8), the weekend effect thus clearly peaks over the largest cities of the Iberian Peninsula. In 315    addition, a few major highways also depict some weekend effect, although this could also be due to minor urbanization along the highway. International shipping emissions are not affected by any significant weekly variability, as demonstrated by the absence of weekend effect along the major maritime routes around Gibraltar despite high TrC-NO$_2$ levels. A more detailed view of the main agglomerations is given in Fig. 9. The weekend effect is limited to -20 % in Palma de Mallorca and Zaragoza, but reaches -30 to -40 % in the other cities. The unprecedentedly high spatial resolution of TROPOMI allows here to reveal 320    substantial gradients of the weekend effect not only between the agglomerations and their surroundings, but also within the agglomeration itself (remind however that functional urban areas cover a much larger area than the agglomerations alone). In particular, in some agglomerations such as Madrid, Barcelona, Valencia or Granada, the strongest weekend effect is not observed above the city center but rather in specific surrounding areas. At a given location, the weekend effect is expected to be mainly driven by (1) the relative share between weekly-variable local (anthropogenic) NOx emissions and weekly-independent 325    local (anthropogenic and natural) NOx emissions, and (2) the weekly variability of the NO$_2$ background advected from the surrounding cells. This emission-related driver may dominate in Madrid or Barcelona where some suburbs can be affected by a stronger relative contribution of commuting (and possibly industry) into the anthropogenic NOx emissions, up to compensate the lower relative contribution of anthropogenic over total NOx emissions that usually peaks in city centers. Conversely, the transport-related driver may be the most important one in the case of Granada as suggested by the clear west-north-west 330    dominant winds, allowing the maximum weekend effect to extend eastward from the city center, down to an area of more complex orography and relatively lower population density. Therefore, summarizing the intensity of the weekend effect in a given city by one single value as done for instance in Goldberg et al. (2021) over the US or in Stavrakou et al. (2020) over worldwide agglomerations provides a useful first level of information but can hide substantial intra-agglomeration variations, especially in the largest metropolitan areas.

335    Apart from the Iberian Peninsula, a weekend effect can also be observed over French cities (Marseille, Toulouse, Bordeaux), although not as strong as in Iberian Peninsula cities. Over Morocco, only a small and diffuse weekend effect can eventually be distinguished in small areas located south of the Ceuta and Melilla Spanish territories. Surprisingly, although Morocco follows the same working days than Spain (i.e. weekend on Saturday-Sunday), no clear weekend effect is observed over





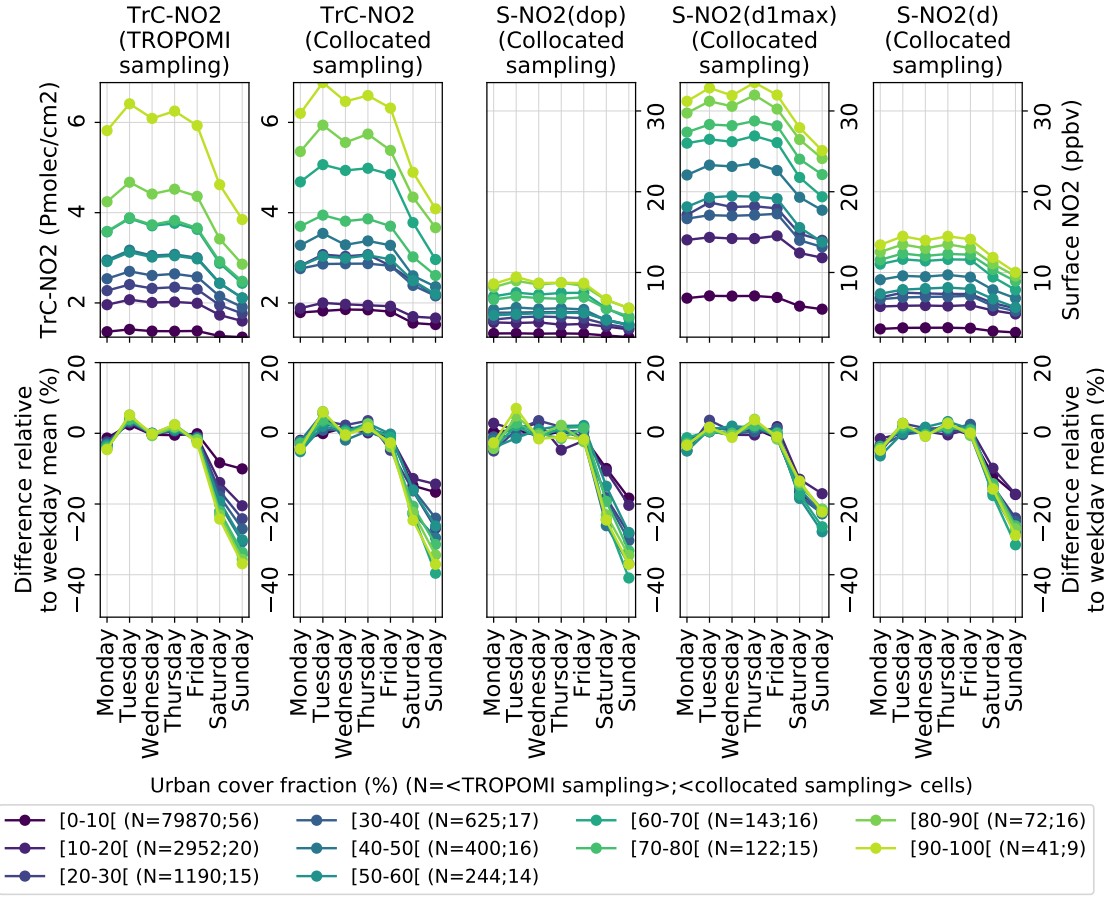

**Figure 6.** Mean weekly profiles (top panels) and differences relative to weekday (Monday-Friday) mean (bottom panels), for different levels of urban cover fraction, considering (from left to right panels): TROPOMI TrC-NO$_2$ full dataset, TROPOMI TrC-NO$_2$ collocated with surface stations, surface NO$_2$ mixing ratios collocated with TROPOMI at daily 24-hour mean (*d*), daily 1-hour maximum (*d1max*) and daily TROPOMI-overpass-time (*dop*) time scales. The collocation of TROPOMI-based and surface-based observations is here performed both spatially and temporally (on a daily basis); the corresponding numbers of cells (averaged over the different days of the week) are indicated in the legend. Urban cover fractions are here binned in bins of 10 %.

Tanger (a relatively large coastal city of Morocco located on the northern point close to Gibraltar), which could be partly due

340   to a substantial contribution of weekly-independent international shipping emissions to ambient NO$_2$ levels. In contrast with Morocco and European countries, Algeria follows a different working days pattern, with Friday-Saturday corresponding to the "weekend", and Friday to the "Sunday", which clearly appears over Algiers and Oran cities when mapping the weekend effect accordingly (see Figs. E2 and E3 in the Appendix). Note that, thanks to the use of traffic congestion data from many countries including some Muslim ones in the updated CAMS-REG-TEMPOv3.2 emission profiles, our estimated NOx anthropogenic





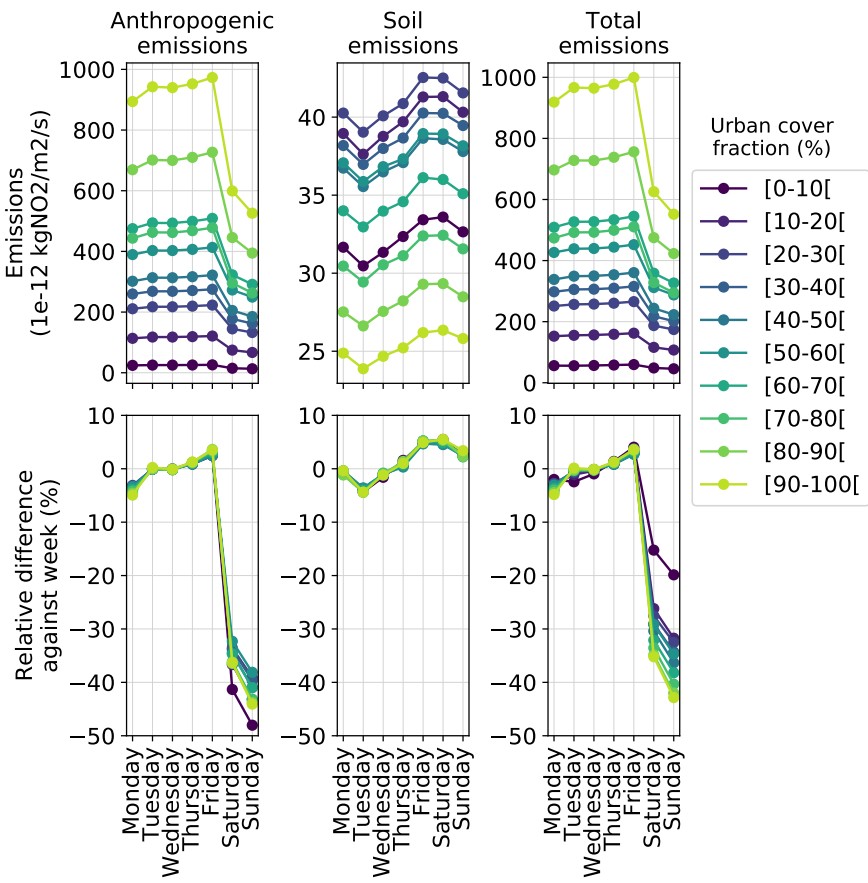

**Figure 7.** Similar to Fig. 6 for daily anthropogenic NOx emissions as obtained with the HERMESv3 model and natural soil NO emissions as calculated with MEGAN (both in $\mathrm{kgNO_2\ m^{-2}\ s^{-1}}$).

emissions in Algiers correctly depict a reduction during Friday-Saturday (-26 and -29 %, respectively), at least qualitatively. However, the weekly profiles used in other important NOx emission sectors such as energy or manufacturing industry do not yet take into account this distribution of the weekdays-weekend, which leads to a persistent reduction on Sunday (-9 %), in disagreement with the TROPOMI TrC-NO$_2$ observations (-35, -15 and -1 % on Friday, Saturday and Sunday, respectively). This illustrates the interest of TROPOMI observations for identifying issues potentially affecting emission temporal profiles in geographical areas where limited activity data information is available.

### 3.4.3 TrC-NO$_2$ weekly variability above industrial sites

We previously mentioned the industry as a potential contributor to the weekly variability of NO$_2$. In order to explore this aspect, we computed the mean weekend reduction at the location of all 5139 industrial point sources emitting NOx in Spain (see Table



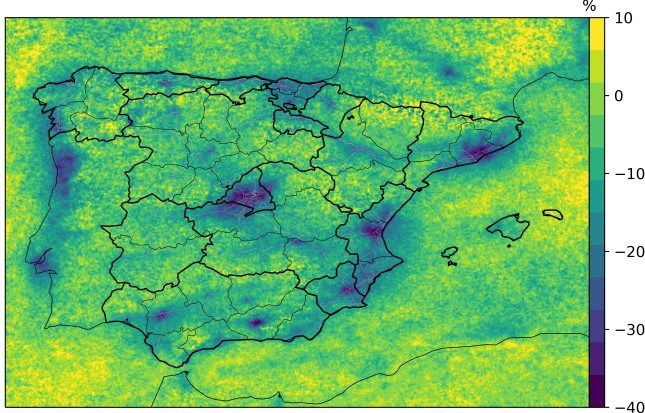

**Figure 8.** Mean relative change of TROPOMI TrC-NO₂ during the weekend (Saturday-Sunday) relative to week (Monday to Friday), over the Iberian Peninsula. Black lines and grey lines correspond to administrative borders and Spanish major roads, respectively (sources: see Sect. 2.4).

E3 in the Appendix). Although results cannot be attributed solely to industry since part of these factories are surrounded by urban area and/or are located downwind, TrC-NO₂ values over these point sources are on average 20 % and 31 % lower on Saturday and Sunday, respectively (25 % over the entire weekend), which would be consistent with the expected decrease of activity during the weekend in most industrial sectors. Interestingly, this mean TrC-NO2 weekend effect progressively decreases when focusing on largest industrial point sources, down to -20 % for the $10^{th}$ largest NOx emitters (-16 % and -24 % on Saturday and Sunday, respectively). Even lower (down to -15%) weekend effects are observed above heavy industries like cement plants, power plants or refineries.

### 3.4.4 Weekly variability of NO₂ in surface monitoring stations

The weekly variability derived from the surface NO₂ measurements also clearly highlights a reduction during the weekend (Fig. 6; third to fifth column panels). Due to diurnal variability of NO₂ pollution, these weekly profiles can change substantially depending on the time scale considered. Depending on the urban cover fraction, the relative reduction of daily mean (*d*) NO₂ ranges between -10 and -20% on Saturday, and between -18 and -32% on Sunday. At daily 1-h maximum (*d1max*) time scale, the reduction tends to be slightly higher on Saturday and slightly lower on Sunday.

In order to compare these results with TROPOMI, the TrC-NO₂ weekly profiles are calculated considering only TROPOMI pixels collocated in time and space with surface NO₂ observations (Fig. 6, second column panels). Compared to the weekly profiles obtained with full TROPOMI sampling, the weekly profiles obtained with the so-called collocated sampling are found to be relatively similar. One difference is the slightly overestimated reduction of TrC-NO₂ during the weekend in least urbanized areas. Also, the relationship between the urban cover fraction and the TrC-NO₂ weekend effect is a bit more noisy compared to the results obtained over the entire domain, which could be a consequence of the much lower number of points.





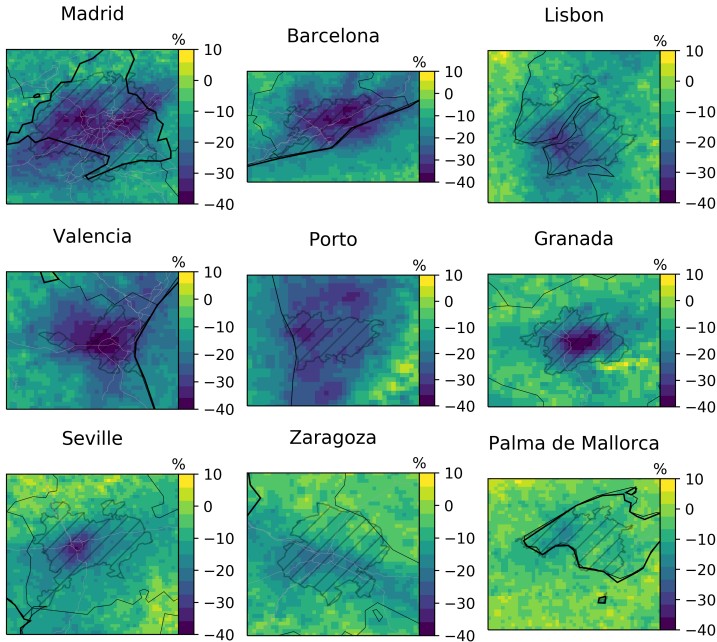

**Figure 9.** Mean relative change of TROPOMI TrC-NO$_2$ during the weekend relative to week, over main Iberian Peninsula cities. Black lines, grey lines and hatched areas correspond to administrative borders, Spanish major roads and functional urban areas, respectively (sources: see Sect. 2.4). Pixels are shown at their 0.025°x0.025° resolution.

The consistency of the weekly variability between TROPOMI TrC-NO$_2$ and surface NO$_2$ is typically better when considering NO$_2$ surface mixing ratios at the hour of TROPOMI overpasses (*dop*), with reductions reaching -40% on Sunday in urbanized areas. In least urbanized areas, surface NO$_2$ show larger differences between Saturday and Sunday compared to TROPOMI TrC-NO$_2$. Among weekdays, both surface NO$_2$ and TROPOMI TrC-NO$_2$ also show relatively consistent variations, but again with slightly stronger discrepancies over least urbanized areas.

### 3.4.5 Comparison with previous studies

Based on OMI (TROPOMI) observations during the period 2005-2017 (05/2018-04/2019), Stavrakou et al. (2020) found Saturday/Sunday relative decreases of -11/-32 % (-9/-30 %) in Barcelona, -0/-18 % (-8/-17 %) in Cordoba, -22/-36 % (-8/-34 %) in Madrid, +13/-10 % (-9/-25 %) in Valencia. Stavrakou et al. (2020) mentioned a flattening of the NO$_2$ weekday-weekend variability over the North America, Europe and Japan due to (1) a decrease of the relative contribution of anthropogenic NOx emissions to total NOx emissions, and (2) an increase of NO$_2$ chemical lifetime driven by the reduction of total NOx emissions and the subsequent decrease of OH levels. Over the United-States, this flattening of the weekend reduction is supported by the recent TROPOMI-based study of Goldberg et al. (2021). For the specific case of the Iberian Peninsula, we investigated the





long-term evolution of the weekend effect as observed at the surface, considering both urban-background and traffic stations (here taken individually, not gridded as in the rest of our study). Results are given in Table E2 in the Appendix. Over the period 1990-2021, the mean annual surface $NO_2$ mixing ratios during the weekend have indeed strongly decreased over Spain and Portugal, from about 10-40 ppbv in the 1990s (but fewer stations) to 10-20 ppbv in the 2000s and 5-10 ppbv over the most

recent years. Nonetheless, even discarding the 1990s when results are less robust due to a limited number of stations, rather than a decrease, results highlight an increase of the weekend effect over the last two decades, although it should be emphasized that a substantial inter-annual variability is also affecting these results.

### 3.5   Monthly variability

#### 3.5.1   TROPOMI-based TrC-$NO_2$ monthly variability against urban land cover fraction

The monthly profiles of TROPOMI TrC-$NO_2$ and surface in-situ $NO_2$ observations are shown in Fig. 10. As for the weekly variability, TrC-$NO_2$ show a stronger monthly variability over most urbanized areas, with larger and lower values during winter and summer, respectively. When the urban cover fraction exceeds 90 %, the differences relative to the annual mean reach +50 % during winter and -40 % during summer. Over the least urbanized areas, they remain below $\pm 20$ %. Again, a consistent picture is obtained when focusing only on the few cells with available surface observations (Fig. 10, second column panels),

except over least urbanized cells where the seasonal amplitude is slightly higher (but these specific profiles are expected to be less comparable given the very different number of accounted cells, 52 against 79,852).

#### 3.5.2   Monthly variability in surface $NO_2$ from monitoring stations

Surface $NO_2$ observations over most urbanized areas also show higher values in winter and lower values in summer. A much larger amplitude of this monthly profile is found when considering surface $NO_2$ at TROPOMI overpass times compared to

daily mean or daily 1-h maximum. TROPOMI TrC-$NO_2$ and surface $NO_2$ at TROPOMI overpass times are very consistent over the least urbanized areas (from -25 % in summer to +30 % in winter). Over the more urbanized areas, the amplitude of the surface $NO_2$ monthly profile is also in general agreement with TROPOMI, although slightly higher. The most noticeable differences concern the overall shape of the monthly profile since surface observations show a broad flat minimum during April-August (against June-August in TROPOMI) and a sharp maximum in December-January (against a relatively broad

maximum in October-February in TROPOMI). Besides potential limitations in terms of robustness (due to the relatively low number of cells taken into account in these comparisons), these discrepancies could originate from monthly variations of the representativeness of the surface observation over the TROPOMI pixel area, but could also reflect some monthly variability in the (normalized) $NO_2$ vertical distribution.

#### 3.5.3   Case study of Madrid for illustrating the flattening of the $NO_2$ pollution seasonal peak

Focusing on the summertime season, such a broad flat minimum over most urbanized areas was not expected given that road transport in many cities is known to be substantially reduced in August when many people go on holidays, although this can be




at least partly compensated by the arrival of national and international tourists, especially on the Spanish coast. As an illustration, we investigated more deeply the case of Madrid. According to ETR and FRONTUR information, the net annual movement of Spanish residents toward Madrid region over 2015-2021 is always minimum in August (-38,735,640 person.d on average),

only slightly compensated by the arrival of international tourists (around +2,362,590 person.d, considering a mean duration of 6 d). This typically induces a clear drop of traffic intensity during August in Madrid, at least in the pre-pandemic era (Table 4, bottom part); traffic counts in Madrid are freely available on the Madrid open data portal (https://datos.madrid.es/portal/site/egob/ menuitem.c05c1f754a33a9fbe4b2e4b284f1a5a0/?vgnextoid=fabbf3e1de124610VgnVCM2000001f4a900aRCRD&vgnextchannel= 374512b9ace9f310VgnVCM100000171f5a0aRCRD&vgnextfmt=default, last access: 01/08/2022). Combined with the fact

that road transport remains the dominant NOx emission sector in Madrid (42 % of total NOx emissions in 2019, in constant decrease over the last decades, https://www.madrid.es/UnidadesDescentralizadas/Sostenibilidad/EspeInf/EnergiayCC/ 04CambioClimatico/4aInventario/Ficheros/Inventario_EAM2019_acc.pdf, last access: 01/08/2022), this naturally implied a minimum of surface $NO_2$ concentrations typically in August, as shown until 2017 (Table 4, top part). However, starting from 2018 (therefore, before the COVID-19 outbreak), the picture starts to be more ambiguous and variable, with some de-

correlations appearing between traffic intensity monthly anomalies and surface $NO_2$ changes. Indeed, apart from the specific year of 2020 strongly impacted by the COVID-19 pandemic (giving a minimum surface $NO_2$ of -56 % in April), surface $NO_2$ monthly levels reached their minimum in June or May over 2018-2021. Besides the continuously decreasing contribution of road transport to total NOx emissions, reasons for these disagreements with the traffic counting observations may include some impact (and/or misrepresentation) of other NOx emission sources with distinct monthly variability and/or some changes in the

chemical and meteorological conditions prevailing in the Madrid region.

### 3.5.4   TrC-NO$_2$ monthly variability over cropland

As previously mentioned, besides anthropogenic sources, soils are another well-known source of NO emissions (Butterbach-Bahl et al., 2013), especially over agricultural areas due to the application of fertilisers (Skiba et al., 2020). Monthly profiles of TrC-NO$_2$ are given in Fig. 11 for different levels of crops cover fraction. Interestingly, besides the expected larger values

observed during cold months, a moderate but noticeable enhancement of TrC-NO$_2$ becomes more visible in June-July as the crop cover fraction increases. We hypothesize that this increase is due to soil emissions, which are known to be very active in summer (Wang et al., 2021). This shows how TROPOMI can provides extremely valuable information given that very few surface $NO_2$ monitoring stations exist in agricultural areas where this enhancement is likely significant; Fig. 11 (right panel) shows the lack of stations over areas with crop cover fraction above 70 % (for reference, the slight summer increase in the

TROPOMI data starts to be visible from a crop cover fraction of 60-70 %).

### 4   Discussion and conclusions

In this study, we comprehensively analysed the TROPOMI TrC-NO$_2$ observations over the Iberian Peninsula, using the recently developed PAL product to ensure consistency across our period of study (2018-2021).





**Table 4.** Monthly mean surface NO$_2$ mixing ratios over 2015-2021, on average over individual monitoring stations located within around 10 km from the city center of Madrid (here including both traffic and urban background stations), and month of minimum NO$_2$ levels (with relative difference against the annual mean into parenthesis). In the bottom, we provide the relative differences (against annual mean) of traffic counting in the Madrid region available over 2018-2021.

| Year | Minimum | Monthly mixing ratio (ppbv) | | | | | | | | | | | |
|------|---------|---|---|---|---|---|---|---|---|---|---|---|---|
| | | J | F | M | A | M | J | J | A | S | O | N | D |
| 2015 | August (-35 %) | 36 | 20 | 23 | 16 | 16 | 17 | 18 | 14 | 20 | 23 | 31 | 35 |
| 2016 | August (-26 %) | 23 | 20 | 20 | 18 | 17 | 17 | 17 | 16 | 24 | 27 | 27 | 31 |
| 2017 | August (-33 %) | 29 | 23 | 22 | 17 | 17 | 17 | 18 | 15 | 22 | 29 | 34 | 29 |
| 2018 | June (-25 %) | 26 | 25 | 16 | 17 | 16 | 15 | 15 | 15 | 20 | 22 | 21 | 29 |
| 2019 | May (-35 %) | 30 | 30 | 20 | 15 | 12 | 13 | 14 | 14 | 17 | 22 | 15 | 22 |
| 2020 | April (-56 %) | 26 | 24 | 12 | 7 | 7 | 9 | 11 | 12 | 15 | 16 | 22 | 17 |
| 2021 | May (-34 %) | 23 | 17 | 16 | 13 | 10 | 11 | 12 | 11 | 15 | 21 | 20 | 22 |
| Year | Minimum | Difference of traffic counting relative to annual mean (%) | | | | | | | | | | | |
| | | J | F | M | A | M | J | J | A | S | O | N | D |
| 2018 | August (-23 %) | 4 | -1 | 0 | 2 | 6 | 6 | -1 | -23 | 0 | 6 | 1 | -2 |
| 2019 | August (-17 %) | -3 | 4 | 0 | 5 | -2 | -3 | -1 | -17 | -1 | 14 | 3 | 1 |
| 2020 | April (-69 %) | 43 | 37 | -18 | -69 | -44 | -1 | 14 | -14 | 14 | 10 | 12 | 15 |
| 2021 | January (-30 %) | -30 | -12 | 1 | -1 | 11 | -3 | 1 | -19 | 10 | 15 | 12 | 13 |

The potential of TROPOMI for supporting the monitoring of NO$_2$ pollution primarily depends on its ability to provide a sufficiently large amount of observations, which is intrinsically related to the cloud coverage affecting the Peninsula. Over 2018-2021, the data availability was estimated to range between 30 and 80 % depending on the season and location. Importantly, it was found to be high enough for providing space-based information on NO$_2$ pollution during 66 % of the days and locations where the NO$_2$ target threshold is exceeded; this coverage increases up to 80-100 % for high O$_3$ episodes in which local NOx play a key role. On a longer-term perspective and for monitoring purposes, it appears important to watch the evolution of the data availability and its impact of climatological TrC-NO$_2$, which appears especially important over Spain where decreasing trends of total cloud cover were reported by Sanchez-Lorenzo et al. (2012) and Sanchez-Lorenzo et al. (2017) since the 1960s (despite an increasing trend in fall).

Analysed at both regional and intra-urban scales, the climatological distribution of TrC-NO$_2$ over the Iberian Peninsula highlights strongest pollution hotspots over Madrid, closely followed by Barcelona (around 7-9 Pmolec cm$^{-2}$), while the other cities typically show lower TrC-NO$_2$ levels (below 5 Pmolec cm$^{-2}$). As an interesting case study, TROPOMI observations were able to capture an extreme and very atypical pollution episode in Madrid during the Filomena snowstorm predominantly attributed to residential heating NOx emissions combined with very low dispersive conditions during several days, in the ab-



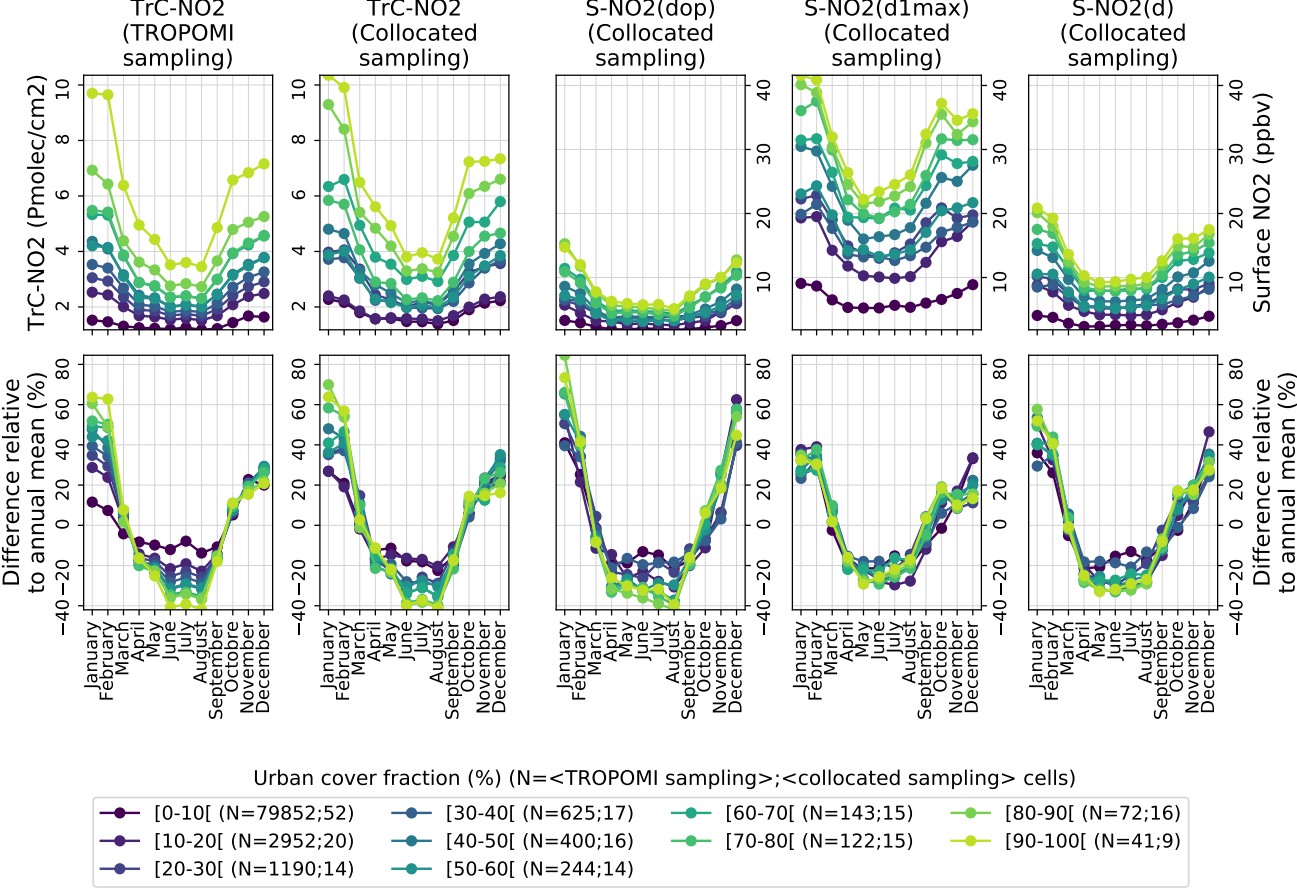

**Figure 10.** Mean monthly profiles (top panels) and differences relative to annual mean (bottom panels), for different levels of urban cover fraction, considering (from left to right panels): TROPOMI TrC-NO$_2$ full dataset, TROPOMI TrC-NO$_2$ collocated with surface stations, surface NO$_2$ mixing ratios collocated with TROPOMI at daily 24-hour mean (*d*), daily 1-hour maximum (*d1max*) and daily TROPOMI-overpass-time (*dop*) time scales. The collocation of TROPOMI-based and surface-based observations is here performed both spatially and temporally (on a daily basis); the corresponding numbers of cells (averaged over the different months of the year) are indicated in the legend. Urban cover fractions are here binned in bins of 10 %.

sence of significant traffic emissions (yet the dominant source in the city). Besides urban areas that are typically covered by at least a few surface monitoring stations, the strongest potential of TROPOMI likely lies in its observations over rural areas and

seas. This appears especially important for the Iberian Peninsula due to an atypically low population density (and thus very scarce surface monitoring network) in a large part of the interior lands excepting Madrid - a phenomenon typically referred to as the "empty Spain" (Llorent-Bedmar et al., 2021) - and the conversely high population density on the coast, often close to major international shipping routes relating the Atlantic Ocean to the Mediterranean sea where intense NOx emissions ad-





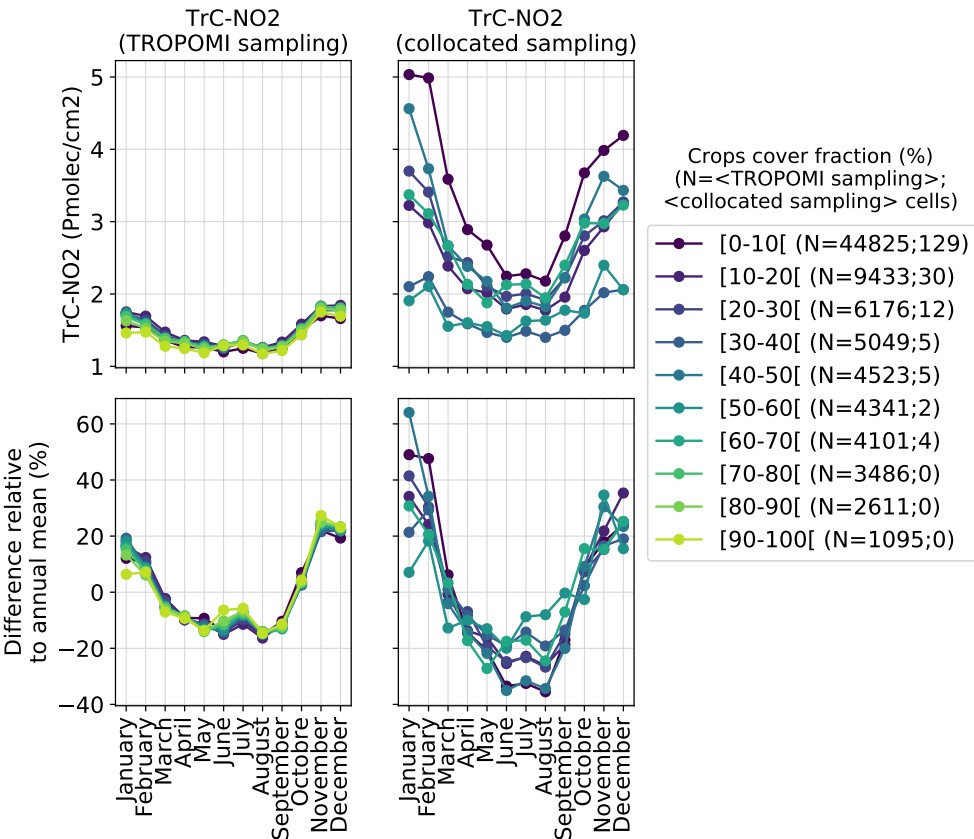

**Figure 11.** Mean monthly profiles (top panels) and differences relative to annual mean (bottom panels), for different levels of crops cover fraction, considering: TROPOMI TrC-NO$_2$ full dataset (left panels) and TROPOMI TrC-NO$_2$ collocated with surface stations (right panels). The collocation of TROPOMI-based and surface-based observations is here performed both spatially and temporally (on a daily basis); the corresponding numbers of cells (averaged over the different months of the year) are indicated in the legend. Crops cover fractions are here binned in bins of 10 %.

versely impact the nearby populations, notably through the production of tropospheric O$_3$. More specifically in rural areas, we
highlighted the potential of TROPOMI for detecting the still highly uncertain natural soil NO emissions prevailing notably over
agricultural areas. In these NOx-limited regions where no surface stations exist, TROPOMI offers an outstanding source of
information for supporting the chemistry-transport modeling efforts and ultimately better characterizing this emission source
of potentially strong impact on O$_3$ production (Lupaşcu and Butler, 2019; Lu et al., 2021).

We characterized in detail the weekly variability of TrC-NO$_2$ and its link with the level of urbanization, highlighting a clear
flattening of the weekend NO$_2$ reduction (relative to weekdays) from -30 to -40 % over most urbanized areas to -10 % over





least urbanized areas. We attributed this persistent small weekend effect (on average) over rural areas to the short- and medium-range transport from the NO$_2$ hotpots and to a small weekly variability in the (low) total (anthropogenic and natural soil) NOx emissions. Focusing on a set of large agglomerations, we found that the weekend effect does not systematically occur in the city center but can in some cases (e.g., Madrid, Barcelona) be observed in some specific suburbs, which could reflect a stronger

relative contribution of commuting to total NOx anthropogenic emissions. Again, such discoveries at intra-urban scale are made possible by the high spatial resolution of TROPOMI, and should support the development and validation of both emission inventories and chemistry-transport models. More specifically, we showed the potential of TROPOMI for supporting the development and evaluation of the emission profiles used to disaggregate the annual anthropogenic emissions provided by emission inventories. However, using TROPOMI observations to better constrain the weekly disaggregation profiles requires

additional information (or assumptions) to take into account the differences of diurnal variability between weekdays and weekend. Indeed, polar-orbiting satellites like S5P with at best one single overpass every day unfortunately cannot provide relevant information, at least over the Iberian Peninsula. Over higher-latitude regions, several S5P overpasses overlap every day, which could in principle provide more insights on the diurnal variability of NO$_2$, but other specific limitations arise (e.g., more frequent presence of snow and clouds, high solar zenith angles, short daylight duration during a large part of the year) (Schneider

et al., 2021). Geostationary missions such as Sentinel 4, TEMPO and GEMS are expected to provide more useful information at the diurnal scale.

In terms of monthly variability, we highlighted a clear increase of the seasonal amplitude in most urbanized areas. We also discovered that the relatively sharp minimum NO$_2$ that was typically occurring in August has evolved over the most recent years toward a broader minimum spanning from late spring to late summer (this change starting in 2018, before some additional

variability was introduced by the COVID and post-COVID situation). Despite an overall consistency, we highlighted some specific discrepancies between surface NO$_2$ and TROPOMI TrC-NO$_2$, especially during spring and fall. Although not yet clearly understood, this demonstrates the importance of analysing jointly space-based and surface-based observations whenever possible, notably because this may provide additional insights on the vertical distribution of NO$_2$, which remains very poorly constrained due to the critical lack of routine airborne observations; this might change in the near-future with the planned

In-service Aircraft for a Global Observing System (IAGOS-CORE) routine NOx measurements on-board commercial aircraft (Berkes et al., 2018).

Following this observation-based study, the important next step is to assess the ability of state-of-the-art chemistry-transport models fed with most recent anthropogenic and natural emissions to reproduce the spatio-temporal variability of TrC-NO$_2$ over the Iberian Peninsula, including its dependency on urban and crops cover fraction. Douros et al. (2022) recently highlighted

some substantial discrepancies between the CAMS regional air quality ensemble and TROPOMI TrC-NO$_2$, especially during wintertime, which could nonetheless points toward issues in both models and TROPOMI retrievals. Considering the aforementioned lack of surface monitoring stations in agricultural areas but potentially strong impact of soil NO emissions on the production of tropospheric O$_3$, evaluating CTMs against TROPOMI in these specific areas is of particular interest, although this likely comes with some challenges due to the relatively weak and diffuse signal of this specific emission source compared

to other anthropogenic sources. Finally, another important aspect to evaluate is the temporal disaggregation currently used





to distribute annual anthropogenic emissions at monthly and daily scale. Although it is often not possible to isolate specific emission sectors, a joint analysis with CTMs and TROPOMI observations can help identifying most critical deficiencies and therefore guide the future efforts for improving these temporal profiles, especially in the sectors (e.g. manufacturing industry, agriculture) where only limited information is available on the temporal variability.

*Data availability.* Sentinel-5p RPRO+OFFL TrC-NO$_2$ data are freely available on the Sentinel-5P Pre-Operations Data Hub (https://scihub. copernicus.eu/, https://doi.org/10.5270/S5P-s4ljg54 for v1, https://doi.org/10.5270/S5P-9bnp8q8 for v2; last access : 10/03/2022). Sentinel-5P PAL TrC-NO$_2$ data are freely available on the S5P-PAL data portal (https://data-portal.s5p-pal.com/; last access : 21/06/2022).

## Appendix A: Comparison between PAL and OFFL+RPRO

In this section, we compare PAL and OFFL+RPRO TROPOMI TrC-NO$_2$ regridded products over their overlapping period
01/05/2018-14/11/2021, focusing on the Iberian Peninsula domain. Taking arbitrarily OFFL+RPRO as the reference, we computed the absolute and normalized mean bias (MB and nMB), the absolute and normalized root mean square error (RMSE and nRMSE), the Pearson Correlation Coefficient (PCC) and the PAL-versus-OFFL+RPRO linear regression slope (slope) (the formulas of these metrics are given in Sect. B in the Appendix). Results are summarized in Table A1.

Overall, both datasets depict a very consistent spatial and temporal variability, with PCC above 0.98. As expected from
its official documentation (https://data-portal.s5p-pal.com/product-docs/no2/PAL_reprocessing_NO2_v02.03.01_20211215. pdf), PAL is showing slightly higher TrC-NO$_2$ values than OFFL+RPRO, with normalized mean differences around +2 %. The PAL-versus-OFFL+RPRO linear regression slopes ranges between 1.06 and 1.09 depending on the time scale. Therefore, the positive increment of TrC-NO$_2$ in PAL tends to be stronger under high TrC-NO$_2$ conditions. However, it is worth noting that this increment does not strictly depend on these TrC-NO$_2$ levels, as illustrated by the variability found among different
cities, with mean relative differences between PAL and OFFL+RPRO reaching +15, +7, +5 and +5 % in the city center of Barcelona, Porto, Madrid (yet the most polluted city) and Lisbon, respectively. The root mean square differences range between 4 % at climatological scale (i.e. when comparing the mean TrC-NO$_2$ maps) and 13 % at daily scale. Consistently with the PAL documentation, the differences between both dataset substantially depends on the season, with larger differences in winter (nMB, nRMSE and slope of +5 %, 18 % and 1.08, respectively) than in summer (nMB, nRMSE and slope of +1 %, 7 %
and 1.02). Note that according to the documentation, the differences are much larger over polluted areas such as China (5-10 % in summer, up to 20-50 % in winter). These differences are notably due to an improved representation of the cloud properties (cloud pressure and cloud radiance fraction) in the FRESCO cloud processor (the mean cloud pressure being substantially overestimated in OFFL+RPRO) (van Geffen et al., 2022b; Compernolle et al., 2021).

As mentioned in Sect. 2.1, the TROPOMI TrC-NO$_2$ dataset used in the rest of the study is composed of the PAL products
until 14/11/2021 combined with the OFFL products after 15/11/2021, and can thus be seen as fully consistent over the period 2018-2021.



**Table A1.** Statistics of TROPOMI TrC-NO2 PAL products, compared to OFFL+RPRO (taken here arbitrarily as the reference).

| Timescale | MB (Pmolec cm$^{-2}$) | nMB (%) | RMSE (Pmolec cm$^{-2}$) | nRMSE (%) | PCC (unitless) | slope (unitless) | N ( points) |
|---|---|---|---|---|---|---|---|
| Climatology | 0.03 | 2.22 | 0.06 | 4.20 | 1.00 | 1.09 | 240,000 |
| Yearly | 0.03 | 2.26 | 0.07 | 5.09 | 1.00 | 1.09 | 960,000 |
| Monthly | 0.03 | 2.27 | 0.13 | 8.85 | 0.99 | 1.09 | 10,298,030 |
| Weekly | 0.03 | 2.36 | 0.18 | 12.51 | 0.98 | 1.07 | 42,505,290 |
| Daily | 0.03 | 1.93 | 0.18 | 12.94 | 0.99 | 1.06 | 185,621,713 |
| Daily (DJF) | 0.08 | 4.58 | 0.32 | 18.46 | 0.98 | 1.08 | 33,689,387 |
| Daily (MAM) | 0.01 | 0.82 | 0.15 | 11.21 | 0.99 | 1.04 | 38,924,612 |
| Daily (JJA) | 0.01 | 0.67 | 0.09 | 7.07 | 0.99 | 1.02 | 64,204,638 |
| Daily (SON) | 0.03 | 1.98 | 0.17 | 11.51 | 0.99 | 1.04 | 48,803,076 |

## Appendix B: Statistical metrics

The statistical metrics used in this study are defined as followed :

$$\text{MB} = \frac{1}{N}\sum_{i=1}^{N} m_i - o_i \tag{B1a}$$

$$\text{nMB} = \frac{\text{MB}}{\overline{o}} \tag{B1b}$$

$$\text{RMSE} = \sqrt{\frac{\sum_{i=1}^{N}(m_i - o_i)^2}{N}} \tag{B1c}$$

$$\text{nRMSE} = \frac{\text{RMSE}}{\overline{o}} \tag{B1d}$$

$$\text{PCC} = \frac{1}{N-1}\sum_{i=1}^{N}\frac{(m_i - \overline{m})(o_i - \overline{o})}{\sigma_m \sigma_o} \tag{B1e}$$

with $m_i$ and $o_i$ the predicted and observed mixing ratios, $\overline{m}$ and $\overline{m}$ their corresponding mean, $\sigma_m$ and $\sigma_m$ their corresponding

standard deviation, and $N$ the number of points.

## Appendix C: Quality assurance with GHOST

Using the metadata available in GHOST (Globally Harmonised Observational Surface Treatment), a quality assurance screening is applied to NO$_2$ hourly observations. A description of the GHOST quality assurance flags used here is given in Table C1.



**Table C1.** Description of the GHOST quality-assurance flags used on the EEA air quality observational dataset.

| Flag | Description |
|---|---|
| 0 | Measurement is missing (i.e. NaN). |
| 1 | Value is infinite – occurs when data values are outside of the range that *float32* data type can handle (-3.4E+38 to +3.4E+38). |
| 2 | Measurement is negative in absolute terms. |
| 3 | Measurement is equal to zero. |
| 6 | Measurements are associated with data quality flags given by the data provider which have been decreed by the GHOST project architects as being associated with substantial uncertainty/bias. |
| 8 | After screening by key QA flags, no valid data remains to average in the temporal window. |
| 10 | The measurement methodology used has not yet been mapped to standardised dictionaries of measurement methodologies. |
| 18 | The specific name of the measurement method is unknown. |
| 20 | The primary sampling is not appropriate to prepare the specific parameter for subsequent measurement. |
| 21 | The sample preparation is not appropriate to prepare the specific parameter for subsequent measurement. |
| 22 | The measurement methodology used is not known to be able to measure the specific parameter. |
| 72 | Measurement is below or equal to the preferential lower limit of detection. |
| 75 | Measurement is above or equal to the preferential upper limit of detection. |
| 82 | The preferential resolution for the measurement is coarser than a set limit (variable by measured parameter). |
| 83 | The resolution of the measurement is analysed month by month. If the minimum difference between observations is coarser than a set limit (variable by measured parameter), measurements are flagged. |
| 90 | Check for persistently recurring values. Check is done by using a moving window of 9 measurements. If 5/6 (i.e. 83.33%) of values in the window are the same then the entire window is flagged. |
| 91 | Check for persistently recurring values. Check is done by using a moving window of 12 measurements. If 9/12 (i.e. 75%) of values in the window are the same, then the entire window is flagged. |
| 92 | Check for persistently recurring values. Check is done by using a moving window of 24 measurements. If 16/24 (i.e. 66.66%) of values in the window are the same, then the entire window is flagged. |
| 110 | The measured value is below or greater than scientifically feasible lower/upper limits (variable by parameter). |
| 111 | The median of the measurements in a month is greater than a scientifically feasible limit (variable by parameter). |
| 112 | Data has been reported to be an outlier through data flags by the network data reporters (and not manually checked and verified as valid). |
| 113 | Data has been found and decreed manually to be an outlier. |
| 131 | 2 out of 3 months' distributions are classed as Zone 6 or higher, suggesting there are potentially systematic reasons for the inconsistent distributions across the 3 months. |
| 132 | 4 out of 6 months' distributions are classed as Zone 6 or higher, suggesting there are potentially systematic reasons for the inconsistent distributions across the 6 months. |
| 133 | 8 out of 12 months' distributions are classed as Zone 6 or higher, suggesting there are potentially systematic reasons for the inconsistent distributions across the 12 months. |

**Appendix D: NO$_2$ pollution in Madrid during the Filomena event**

The historical maximum TrC-NO$_2$ across the Iberian Peninsula reached 63 Pmolec cm$^{-2}$. Interestingly, it occurred in early January 2021 in Madrid a few days after Filomena - the largest snowstorm since 1971 - hit central Spain (Tapiador et al., 2021) (see the day-to-day evolution of TrC-NO$_2$ during this event in Fig. D1). During and right after this extreme event, local



road transport NOx emissions were very low due to an exceptional snow coverage, preventing cars to circulate in business-as-usual conditions. On average over the 7-11/01/2021, traffic count observations over Madrid indeed show a reduction of 70 % compared to the same period in 2019 (Madrid open data portal, https://datos.madrid.es/, last access : 01/08/2022). In spite of that, TROPOMI measured unusually high TrC-$NO_2$ values (pixels with substantial snow coverage are expected to be already filtered with the *qa_value*), which thus appears to be explained by the accumulation of $NO_2$ pollution originating from other sources, first and foremost residential combustion for heating (mainly with gas in Madrid) in a very stable and shallow boundary layer (whose daily mean height ranges between 42 and 108 m during these days according to ERA5 reanalysis). Similarly, very high surface $NO_2$ mixing ratios were observed by surface stations during this event, with daily mean values often exceeding the $99.9^{th}$ percentile (48 ppbv) over all surface $NO_2$ observations over the domain and period of study. As this high TrC-$NO_2$ was measured in pixels close to (filtered) cloudy pixels, it is also worth mentioning here that it might be affected by some biases related to cloud shadowing effects. Indeed, recent studies on the 3-dimensional cloud structure and the corresponding impact of shadowing on $NO_2$ retrievals in pixels neighbouring clouds estimated that biases typically remain below 10 % for low (below 40°) solar zenith angles but can reach tens of percent for larger angles (Yu et al., 2021; Emde et al., 2022; Kylling et al., 2022).



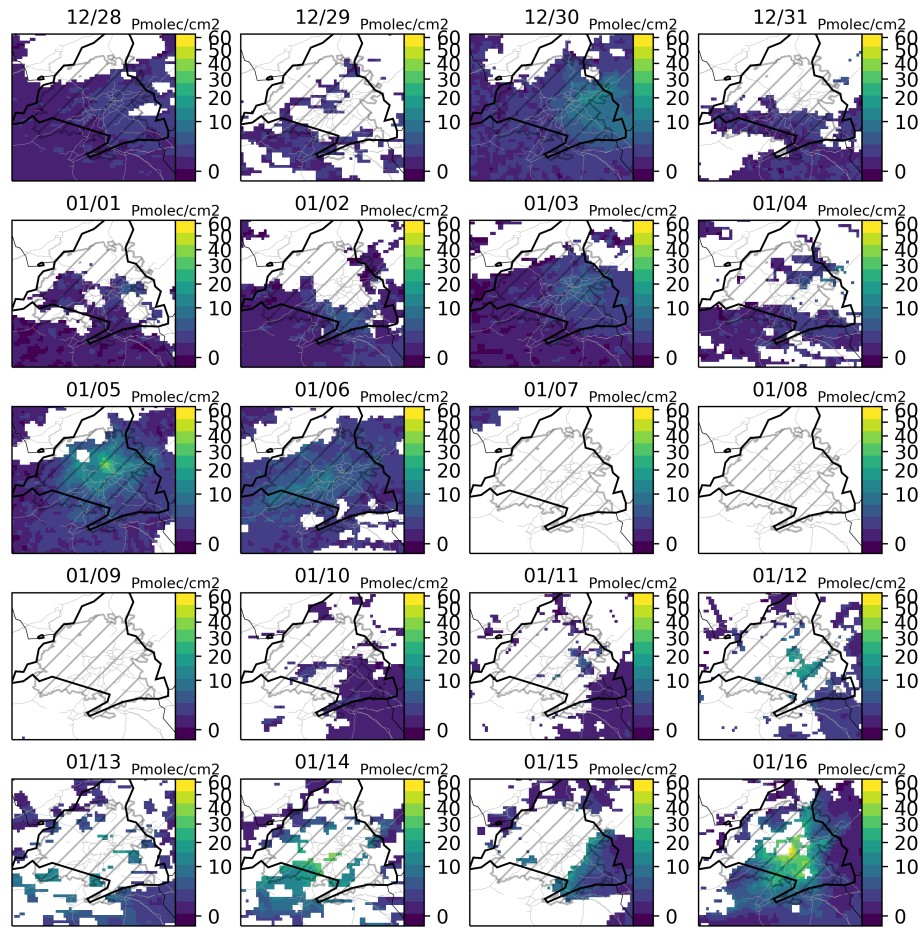

**Figure D1.** TrC-NO$_2$ measured by TROPOMI over Madrid region, during the two weeks preceding the 16 January 2021, day of historical maximum TROPOMI TrC-NO$_2$ over the Iberian Peninsula. The Filomena snowstorm occurred mainly between 7-11/01/2021 (cloudy). Note that the color scale is not linear. Black lines, grey lines and hatched areas correspond to administrative borders, Spanish major roads and functional urban areas, respectively (sources: see Sect. 2.4).





## Appendix E: Other figures and tables

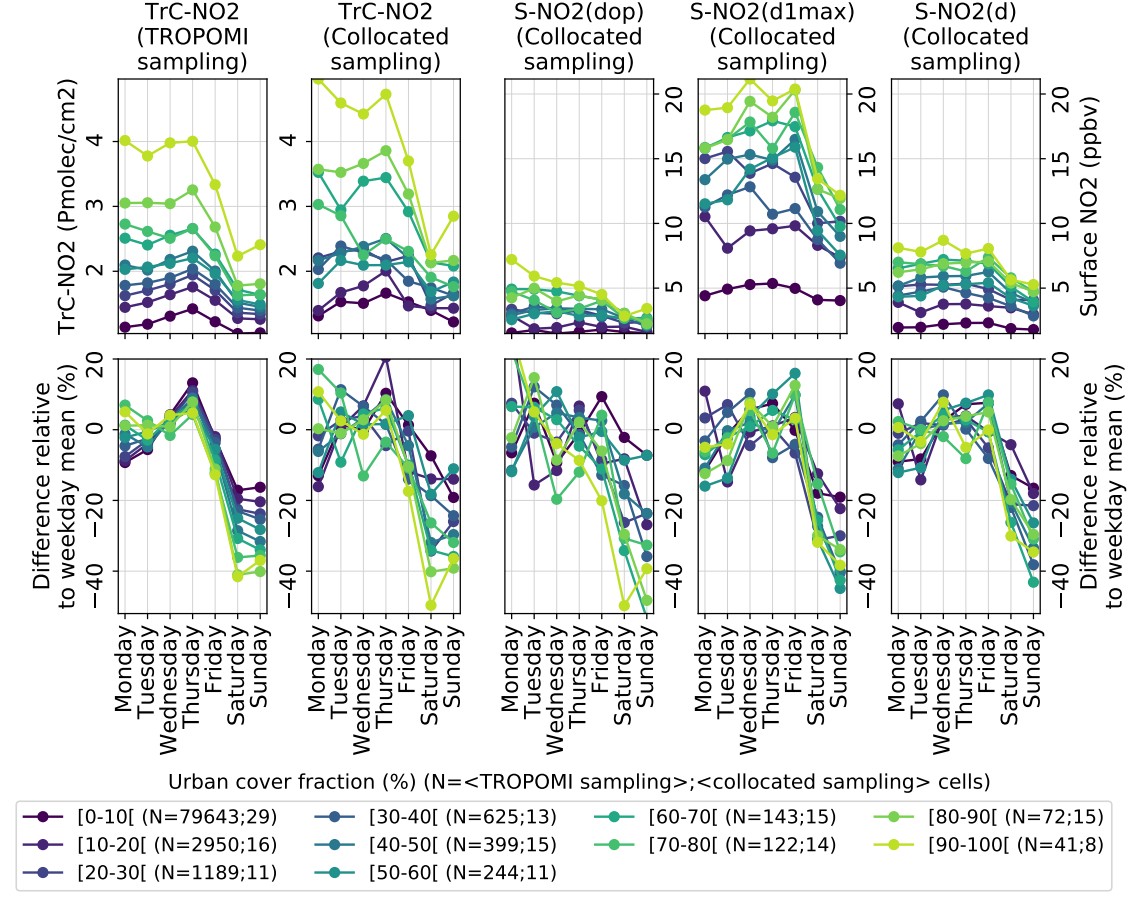

**Figure E1.** Mean weekly profiles (top panels) and differences relative to weekday (Monday-Friday) mean (bottom panels), during the first strict COVID-19 lockdown (15/03/2021-31/05/2021), for different levels of urban cover fraction, considering (from left to right panels): TROPOMI TrC-NO$_2$ full dataset, TROPOMI TrC-NO$_2$ collocated with surface stations, surface NO$_2$ mixing ratios collocated with TROPOMI at daily 24-hour mean (*d*), daily 1-hour maximum (*d1max*) and daily TROPOMI-overpass-time (*dop*) time scales. The collocation of TROPOMI-based and surface-based observations is here performed both spatially and temporally (on a daily basis); the corresponding numbers of cells (averaged over the different days of the week) are indicated in the legend. Urban cover fractions are here binned in bins of 10 %.





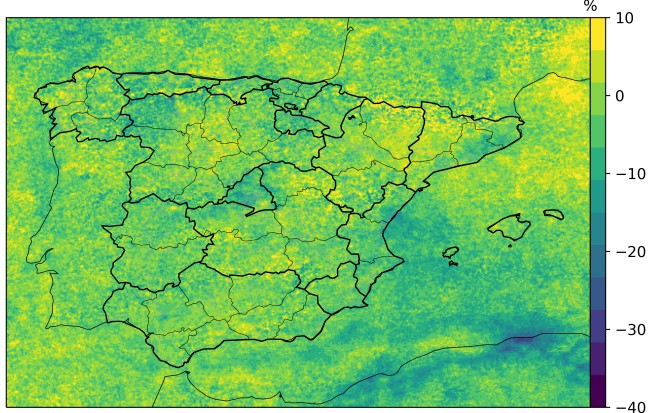

**Figure E2.** Mean relative change of TROPOMI TrC-NO$_2$ on Friday-Saturday (which corresponds to the occidental "weekend" in Algeria), relative to the other days (Monday-Tuesday-Wednesday-Thursday-Sunday). Note that weekends in Morocco are similar to Spain.

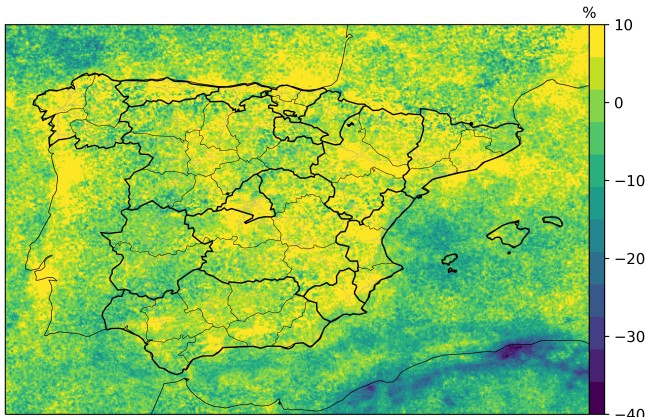

**Figure E3.** Mean relative change of TROPOMI TrC-NO$_2$ on Friday (which corresponds to the occidental "Sunday" in Algeria), relative to the other days (Monday-Tuesday-Wednesday-Thursday-Sunday). Black lines, grey lines and hatched areas correspond to administrative borders, Spanish major roads and functional urban areas, respectively (sources: see Sect. 2.4).





**Table E1.** Mean data availability of TROPOMI TrC-NO$_2$ over the different regions of the Iberian Peninsula, on average over 2018-2021.

| Region | Availability |
|---|---|
| Alentejo region (PT) | 63% |
| Algarve region (PT) | 66% |
| Andalusia (ES) | 67% |
| Aragon (ES) | 56% |
| Cantabria (ES) | 34% |
| Castilla y Leon (ES) | 51% |
| Castilla-La Mancha (ES) | 59% |
| Catalonia (ES) | 53% |
| Center region (PT) | 55% |
| Ceuta city (ES) | 63% |
| Melilla city (ES) | 67% |
| Navarre (ES) | 45% |
| Madrid region (ES) | 59% |
| Valencia (ES) | 62% |
| Extremadura (ES) | 62% |
| Galicia (ES) | 42% |
| Balearic Islands (ES) | 60% |
| La Rioja (ES) | 45% |
| Northern region (PT) | 51% |
| Basque Country (ES) | 38% |
| Asturias (ES) | 34% |
| Murcia (ES) | 68% |
| Lisbon Metropolitan Area (PT) | 61% |



**Table E2.** Annual evolution of the entire weekend (as well as Saturday/Sunday taken individually, into parenthesis) mean surface NO$_2$ mixing ratios and change, relative to weekdays (Monday to Friday), over the period 1990-2021 in Spain.

| Year | Urban background stations | | | Traffic stations | | |
|------|------------|------------------|----------------|------------|------------------|----------------|
| | N(stations) | mean NO$_2$ (ppbv) | weekend effect | N(stations) | mean NO$_2$ (ppbv) | weekend effect |
| 1990 | 0 | - | - | 0 | - | - |
| 1991 | 1 | 39.7 (40.9/38.6) | -8% (-5%/-10%) | 5 | 27.6 (29.3/26.0) | -16% (-11%/-21%) |
| 1992 | 1 | 35.7 (37.0/34.4) | -14% (-11%/-18%) | 5 | 26.7 (27.8/25.7) | -14% (-11%/-18%) |
| 1993 | 2 | 30.5 (31.9/29.0) | -14% (-10%/-18%) | 6 | 27.8 (29.1/26.5) | -13% (-9%/-17%) |
| 1994 | 0 | - | - | 0 | - | - |
| 1995 | 2 | 28.3 (30.0/26.7) | -13% (-8%/-18%) | 6 | 29.2 (30.7/27.8) | -15% (-10%/-19%) |
| 1996 | 2 | 27.5 (28.8/26.2) | -15% (-11%/-19%) | 6 | 29.3 (31.0/27.6) | -13% (-8%/-18%) |
| 1997 | 19 | 13.8 (14.9/12.7) | -18% (-11%/-24%) | 69 | 23.6 (25.0/22.2) | -17% (-12%/-22%) |
| 1998 | 19 | 13.9 (14.8/12.9) | -17% (-11%/-22%) | 72 | 23.3 (25.1/21.5) | -17% (-11%/-24%) |
| 1999 | 35 | 10.5 (11.1/9.9) | -19% (-15%/-24%) | 86 | 21.0 (22.6/19.3) | -22% (-16%/-28%) |
| 2000 | 44 | 9.5 (10.2/8.7) | -17% (-11%/-24%) | 87 | 20.0 (21.7/18.3) | -18% (-11%/-25%) |
| 2001 | 73 | 9.0 (9.5/8.5) | -19% (-15%/-24%) | 105 | 18.8 (20.0/17.5) | -22% (-17%/-27%) |
| 2002 | 83 | 9.1 (9.7/8.5) | -17% (-11%/-23%) | 99 | 18.8 (20.3/17.3) | -19% (-13%/-26%) |
| 2003 | 98 | 8.8 (9.6/8.0) | -21% (-14%/-28%) | 101 | 17.7 (19.3/16.1) | -22% (-14%/-29%) |
| 2004 | 104 | 8.8 (9.4/8.2) | -23% (-17%/-28%) | 98 | 17.1 (18.4/15.7) | -23% (-17%/-29%) |
| 2005 | 146 | 9.2 (10.0/8.5) | -22% (-15%/-28%) | 149 | 19.0 (20.8/17.2) | -21% (-14%/-28%) |
| 2006 | 150 | 8.3 (8.8/7.8) | -23% (-18%/-27%) | 121 | 16.8 (18.0/15.7) | -23% (-18%/-28%) |
| 2007 | 167 | 9.0 (9.6/8.3) | -18% (-13%/-24%) | 127 | 17.4 (18.6/16.1) | -20% (-14%/-26%) |
| 2008 | 174 | 7.6 (8.2/7.1) | -22% (-16%/-28%) | 116 | 15.8 (17.1/14.5) | -21% (-15%/-28%) |
| 2009 | 186 | 7.9 (8.5/7.4) | -21% (-15%/-27%) | 134 | 16.0 (17.3/14.7) | -24% (-18%/-30%) |
| 2010 | 202 | 8.0 (8.5/7.5) | -21% (-17%/-26%) | 127 | 14.6 (15.7/13.4) | -24% (-18%/-30%) |
| 2011 | 210 | 7.7 (8.2/7.2) | -22% (-16%/-27%) | 122 | 14.1 (15.3/12.8) | -24% (-17%/-30%) |
| 2012 | 204 | 7.3 (7.8/6.9) | -22% (-18%/-27%) | 116 | 14.0 (15.1/12.9) | -23% (-17%/-29%) |
| 2013 | 213 | 7.5 (7.9/7.1) | -18% (-14%/-23%) | 109 | 13.0 (13.9/12.1) | -22% (-17%/-27%) |
| 2014 | 219 | 7.2 (7.6/6.7) | -21% (-16%/-26%) | 116 | 12.4 (13.4/11.4) | -24% (-18%/-30%) |
| 2015 | 213 | 7.9 (8.4/7.4) | -19% (-14%/-24%) | 118 | 13.5 (14.5/12.5) | -22% (-16%/-27%) |
| 2016 | 220 | 7.3 (7.8/6.7) | -21% (-16%/-27%) | 114 | 12.3 (13.5/11.1) | -25% (-17%/-32%) |
| 2017 | 227 | 7.7 (8.1/7.2) | -21% (-17%/-26%) | 118 | 13.1 (14.1/12.0) | -23% (-17%/-29%) |
| 2018 | 234 | 6.7 (7.2/6.2) | -23% (-18%/-28%) | 120 | 11.7 (12.6/10.8) | -25% (-19%/-30%) |
| 2019 | 235 | 6.8 (7.3/6.3) | -18% (-12%/-24%) | 122 | 11.7 (12.9/10.6) | -21% (-13%/-28%) |
| 2020 | 244 | 5.5 (5.8/5.1) | -21% (-16%/-26%) | 122 | 9.0 (9.8/8.3) | -25% (-19%/-31%) |
| 2021 | 246 | 5.2 (5.5/4.8) | -24% (-19%/-29%) | 122 | 8.5 (9.2/7.8) | -29% (-23%/-35%) |



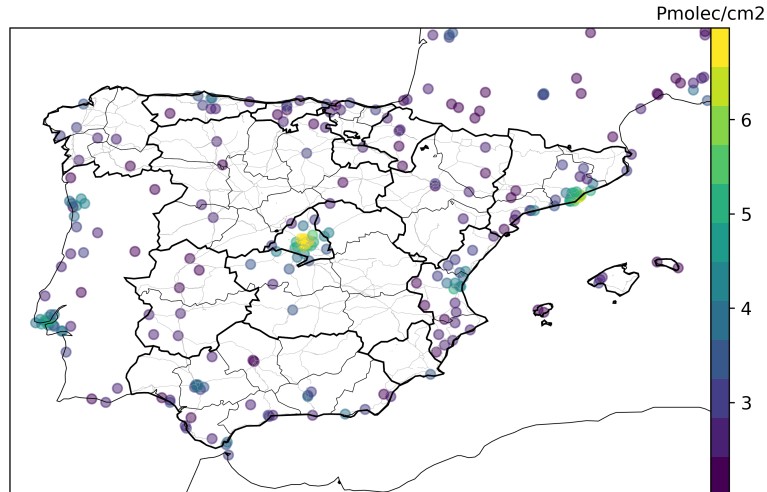

**Figure E4.** Mean TrC-NO$_2$ coincident with the 10 % largest residuals of the daily-scale TrC-NO$_2$ versus surface NO$_2$ linear regression. Black and grey lines correspond to administrative borders and Spanish major roads, respectively (sources: see Sect. 2.4).

**Table E3.** Mean TROPOMI-based TrC-NO$_2$ weekend effect above different groups of industrial point sources across Spain. pX here corresponds to the X$^{\text{th}}$ percentile of the mean emission of the different industrial point sources. Specific industrial activities were grouped according to the Selected Nomenclature for Air Pollution (SNAP) classification system.

| Sub-group | Number of industries | Mean annual emission per point source (kgNOx y$^{-1}$) | Weekend | Saturday | Sunday |
|---|---|---|---|---|---|
| All | 5139 | 41762 | -25 % | -20 % | -31 % |
| All $\geq$ p10 | 5012 | 42820 | -25 % | -20 % | -31 % |
| All $\geq$ p20 | 5012 | 42820 | -25 % | -20 % | -31 % |
| All $\geq$ p30 | 3642 | 58928 | -23 % | -18 % | -28 % |
| All $\geq$ p40 | 3561 | 60268 | -23 % | -18 % | -28 % |
| All $\geq$ p50 | 2570 | 83508 | -22 % | -18 % | -26 % |
| All $\geq$ p60 | 2056 | 104302 | -22 % | -18 % | -26 % |
| All $\geq$ p70 | 1542 | 138472 | -21 % | -17 % | -25 % |
| All $\geq$ p80 | 1028 | 205547 | -21 % | -17 % | -25 % |
| All $\geq$ p90 | 514 | 394694 | -20 % | -16 % | -23 % |
| SNAP030311 (cement plants) | 34 | 758175 | -19 % | -15 % | -23 % |
| SNAP010101 (coal-fired power plants) | 18 | 748058 | -17 % | -14 % | -20 % |
| SNAP010104 (natural gas power plants) | 53 | 208534 | -18 % | -13 % | -22 % |
| SNAP010301 to SNAP010305 (refineries) | 31 | 337638 | -15 % | -12 % | -18 % |



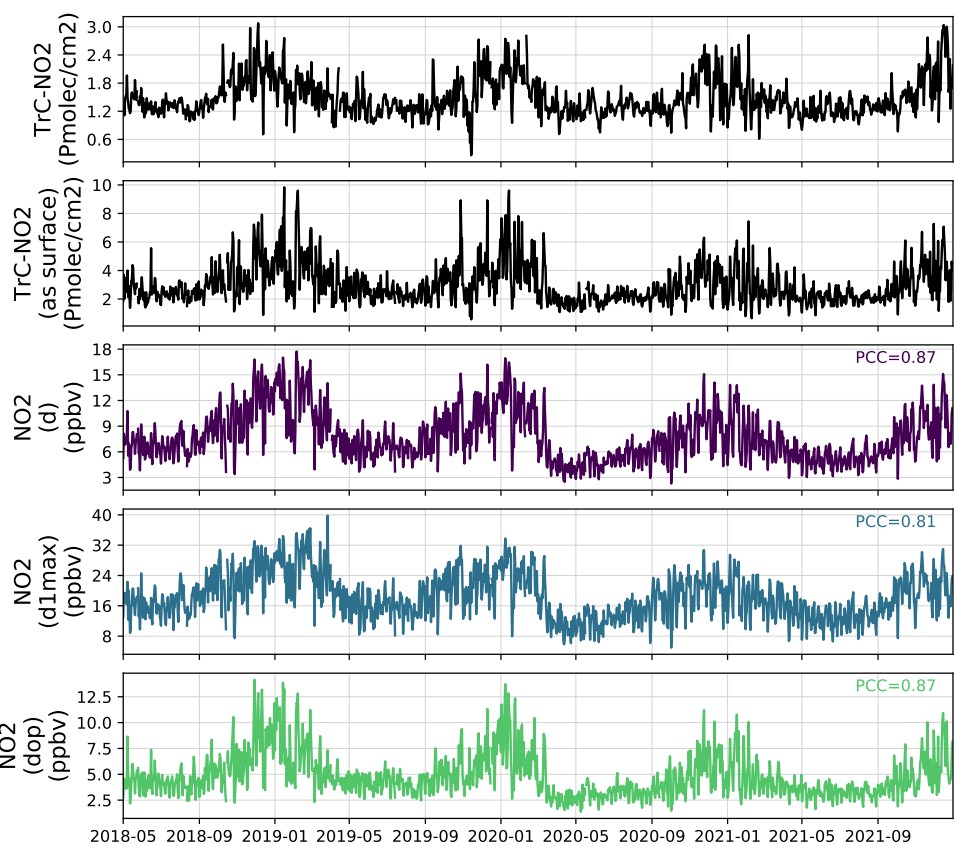

**Figure E5.** Mean daily time series of TROPOMI TrC-NO₂ (over the entire domain, or restricted to cells with surface observations available; top 2 panels) and surface NO₂ mixing ratios, at daily (*d*), daily 1-h maximum (*d1max*) and collocated with daily TROPOMI overpasses (*dop*) (bottom 3 panels). The Pearson Correlation Coefficients (PCC) of surface NO₂ mixing ratios against TROPOMI TrC-NO₂ restricted to cells with surface observations available are indicated.



*Author contributions.* HP designed the study and carried out the analysis. PAB was responsible for downloading the TROPOMI and mete-
orological data. DB was responsible for the acquisition and preprocessing of the air quality data through the GHOST project. MG and SE
were responsible for computing the anthropogenic and natural NOx emissions. HP was responsible for writing the article. MG contributed
especially on the discussion related to emissions. SC reviewed the discussion related to TROPOMI data. HP, CPGP, SC, OJ, AS, MG and FL
contributed to the interpretation of results and the review of the article.

*Competing interests.* The authors declare that they have no conflict of interest.

*Acknowledgements.* This project has received funding from the European Union's Horizon 2020 research and innovation programme un-
der grant agreement No 870301 (AQ-WATCH H2020 project), as well as the MITIGATE project (PID2020-116324RA695 I00 / AEI /
10.13039/501100011033) from the Agencia Estatal de Investigacion (AEI). We also acknowledge support from the VITALISE project
(PID2019-108086RA-I00) funded by MCIN/AEI/10.13039/501100011033, the AXA Research Fund and Red Temática ACTRIS España
(CGL2017-90884-REDT), RES (AECT-2022-1-0008, AECT-2022-2-0003) for awarding us access to Marenostrum Supercomputer in the
Barcelona Supercomputing Center, and H2020 ACTRIS IMP (#871115). SC acknowledges support from BELSPO through BRAIN-BE 2.0
project LEGO-BEL-AQ (contract B2/191/P1/LEGO-BEL-AQ). Last but not least, we gratefully acknowledge the outstanding work done by
the Python development teams behind some specific libraries, including *numpy*, *pandas*, *xarray*, *matplotlib*, *cartopy*, and *xESMF*.



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
