# Peer review of "Potential of TROPOMI for understanding spatio-temporal variations in surface NO2 and their dependencies upon land use over the Iberian Peninsula"

_EGUsphere, 2022_

## Referee Comment (RC1)

This is an excellent manuscript describing TROPOMI NO2 variability and correlation with surface measurements in the Iberian Peninsula. Although I have provided many suggestions, most of them are very minor and should be easy to address. One small weakness is that the writing is a bit long-winded at times, and there are opportunities to shorten. I have noted several examples of text that could benefit from shortening.

I am also surprised that the authors have limited discussion on interannual trends in the paper. We now have 4 years of May - Dec data from TROPOMI. Perhaps something like this could be discussed in Section 3.5? For example, are we seeing lower values in 2021 than 2018? We know the answer is yes for 2020 based on other literature and perhaps that should be mentioned briefly, but how about in 2021? It seems like that's an important aspect that is missing from this paper. This would constitute my one "major" comment.

Minor comments:
Line 23. Remove "all in all"

Line 30. Add the word "fossil-fuel" or "NOx" in front of the word "emissions"

Line 30. Not sure why the word "essentially" is used. Suggest removal. Or add a short phrase about low-cost monitors.

Line 53. Add guideline value (5.3 ppb or 10 ug/m3)

Line 61. Maybe add a few references that discuss uncertainties of NOx inventories in Europe. Pope et al., 2022, Zara et al., 2021, Goldberg et al., 2021 could be cited here.

-Pope, R. J., Kelly, R., Marais, E. A., Graham, A. M., Wilson, C., Harrison, J. J., Moniz, S. J. A., Ghalaieny, M., Arnold, S. R. and Chipperfield, M. P.: Exploiting satellite measurements to explore uncertainties in UK bottom-up NOx emission estimates, Atmos. Chem. Phys., 22(7), 4323–4338, doi:10.5194/ACP-22-4323-2022, 2022.
-Zara, M., Boersma, K. F., Eskes, H. J., Denier van der Gon, H., Vilà-Guerau de Arellano, J., Krol, M., van der Swaluw, E., Schuch, W. and Velders, G. J. M.: Reductions in nitrogen oxides over the Netherlands between 2005 and 2018 observed from space and on the ground: Decreasing emissions and increasing O3 indicate changing NOx chemistry, Atmos. Environ. X, 9, 100104, doi:10.1016/j.aeaoa.2021.100104, 2021.
-Goldberg, D. L., Anenberg, S. C., Lu, Z., Streets, D. G., Lamsal, L. N., E McDuffie, E. and Smith, S. J.: Urban NOx emissions around the world declined faster than anticipated between 2005 and 2019, Environ. Res. Lett., 16(11), 115004, doi:10.1088/1748-9326/ac2c34, 2021.

Lines 80 - 94. Opportunity to make more concise. Mentioning NRTI, RPRO and L1b data are probably unnecessary. For example, mention that PAL is used (one sentence), mention that this is different than the OFFL product (one sentence), mention differences between the two

products (one sentence), and that more details can be found in PUM or ATBD (one sentence). This paragraph could be shortened from 8 sentences to perhaps 4-5.

Lines 108 - 122. Opportunity to make more concise. Current text could be shortened to 2-3 sentences.

Lines 108 - 122. A 1-2 sentence summary of Appendix A and van Geffen et al. 2022 paper should be discussed in this paragraph. Most notably that PAL product yields larger values that the OFFL/RPRO product

Line 155. Mention the word "fertilizer" to be more explicit.

Line 168. No need to mention vertical level information if you are only using near-surface variables

Line 216. Add word "long-term" before the word "values". I believe you are referring to the maximum of the May 2018 - Dec 2021 average, and not the daily maximum.

Figure 1, 2, 3 captions - Please mention that the oversampled images are a May 2018 - Dec 2021 average.

Line 228 - Worth mentioning in this sentence the potential for this pixel to be an artifact of snow/clouds (as discussed in Appendix D)

Line 249. Need 1-2 clarifying sentences inserted here to describe Figure 4, and why it was generated. It's not exactly clear what point you are trying to make with Figure 4. I think Lines 252 - 256 are referring to Figure 4, but this isn't clear.

Line 255. Remove the word "too". Also what is meant by "limited number of points"? I think you mean to say that "averaging reduces the sample size".

Lines 258 - 269. Opportunity to make more concise. This could probably be shortened to 1-3 sentences.

Line 303. I see a weekly cycle in Figure 7, in that soil NOx emissions are largest on Fridays. Is this driven by a fertilizer application cycle? It does not seem that meteorological variability is the cause.

Line 310. I also see a slight uptick on Thursday. It would be interesting to see if the TomTom data also shows upticks on Tuesdays (and Thursdays). Do you have access to any traffic data?

Figure 6. If there's room, if you can change S-NO2 to Surface-NO2 in the top label that may bring more clarity. Maybe this will mean 4 rows of text for the top label instead of 3. In the

figure caption, the order of "d" and "dop" are swtiched. "Top" is listed in the third column but mentioned as the fifth in the caption.

Line 419. What are the units person.d?

Section 3.5.3. What is the main takeaway point of this Section? It is not clear to me. Based on the current text, I would suggest removal of this section, but perhaps I am missing the point.

Figure 10. Same comment as Figure 6. If there's room, if you can change S-NO2 to Surface-NO2 in the top label that may bring more clarity.

Lines 480 - 489. Opportunity to make more concise. These 5 sentences could probably be 2 or 3 sentences instead.

---

## Author Response (AR1)

We thank the three anonymous referees for reading carefully our study and providing useful comments and feedbacks. In this document, referee's comments are in grey, our answers in black and the modifications applied to the manuscript in blue (highlighted in bold, when relevant).

**Answers to referee #1**

This is an excellent manuscript describing TROPOMI NO2 variability and correlation with surface measurements in the Iberian Peninsula. Although I have provided many suggestions, most of them are very minor and should be easy to address. One small weakness is that the writing is a bit long-winded at times, and there are opportunities to shorten. I have noted several examples of text that could benefit from shortening.

We thank the referee for his/her positive feedback on our study. In order to be a bit more concise, we shortened some specific paragraphs and moved some sections in the Appendix, as discussed below.

I am also surprised that the authors have limited discussion on interannual trends in the paper. We now have 4 years of May - Dec data from TROPOMI. Perhaps something like this could be discussed in Section 3.5? For example, are we seeing lower values in 2021 than 2018? We know the answer is yes for 2020 based on other literature and perhaps that should be mentioned briefly, but how about in 2021? It seems like that's an important aspect that is missing from this paper. This would constitute my one "major" comment.

We agree this is an interesting point to mention. We included another small section discussing the inter-annual variability in TROPOMI and surface observations (but focusing on years with full coverage, thus 2019-2021).

Minor comments:
Line 23. Remove "all in all"
Done.

Line 30. Add the word "fossil-fuel" or "NOx" in front of the word "emissions"
Done.

Line 30. Not sure why the word "essentially" is used. Suggest removal. Or add a short phrase about low-cost monitors.
We applied the following modification : "The monitoring of surface NO2 pollution essentially relies on official air quality (AQ) surface stations **(passive and low-cost sensors being two other sources of information, although of lower temporal resolution and poorer quality, respectively)**."

Line 53. Add guideline value (5.3 ppb or 10 ug/m3)
We applied the following modification : "[...] more than 80% of the surface monitoring stations keep reporting NO2 (and O3, on which NOx play a key role) levels well above the guidelines **(10 and 25 µg/m3 on annual and daily average, respectively)** recommended by the World Health Organization [...]."

Line 61. Maybe add a few references that discuss uncertainties of NOx inventories in Europe. Pope et al., 2022, Zara et al., 2021, Goldberg et al., 2021 could be cited here. -Pope, R. J., Kelly, R., Marais, E. A., Graham, A. M., Wilson, C., Harrison, J. J., Moniz, S. J. A., Ghalaieny, M., Arnold, S. R. and Chipperfield, M. P.: Exploiting satellite measurements to explore uncertainties in UK bottom-up NOx emission estimates, Atmos. Chem. Phys., 22(7), 4323–4338, doi:10.5194/ACP-22-4323-2022, 2022. -Zara, M., Boersma, K. F., Eskes, H. J., Denier van der Gon, H., Vilà-Guerau de Arellano, J., Krol, M., van der Swaluw, E., Schuch, W. and Velders, G. J. M.: Reductions in nitrogen oxides over the Netherlands between 2005 and 2018 observed from space and on the ground: Decreasing emissions and increasing O3 indicate changing NOx chemistry, Atmos. Environ. X, 9, 100104, doi:10.1016/j.aeaoa.2021.100104, 2021. -Goldberg, D. L., Anenberg, S. C., Lu, Z., Streets, D. G., Lamsal, L. N., E McDuffie, E. and Smith, S. J.: Urban NOx emissions around the world declined faster than anticipated between 2005 and 2019, Environ. Res. Lett., 16(11), 115004, doi:10.1088/1748-9326/ac2c34, 2021.

Done.

Lines 80 - 94. Opportunity to make more concise. Mentioning NRTI, RPRO and L1b data are probably unnecessary. For example, mention that PAL is used (one sentence), mention that this is different than the OFFL product (one sentence), mention differences between the two products (one sentence), and that more details can be found in PUM or ATBD (one sentence). This paragraph could be shortened from 8 sentences to perhaps 4-5.

Although the paper is focusing on the PAL product, we also perform a brief comparison against OFFL-RPRO so we need to introduce all these products as well. In addition, all readers may not be familiar with these different TROPOMI TrC-NO2 products. Therefore, we think most of the information provided here is useful (especially since TROPOMI is the main data used in our study). However, we made this paragraph more concise :

"So-called reprocessed (RPRO) and offline (OFFL) TrC-NO2 products covering the periods 30/04/2018-17/10/2018 and 17/10/2018-present, respectively, are publicly delivered as L2 products along S5P orbits on the S5P hub (https://scihub.copernicus.eu/, https://doi.org/10.5270/S5P-s4ljg54 for v1, https://doi.org/10.5270/S5P-9bnp8q8 for v2; last access : 10/03/2022); More details about these products can be found in the Algorithm Theoretical Basis Document (Geffen et al., 2022), the Product User Manual (Eskes et al., 2022a) and the Product Readme File (Eskes et al., 2022). In late 2021, the so-called PAL TrC-NO2 product covering the period 01/05/2018-14/11/2021 has been made publicly available (https://data-portal.s5p-pal.com/; last access : 21/06/2022) (Eskes et al., 2022b). Based on the last processor version available at that time (namely the version 2.03.01), PAL offers a consistent TrC-NO2 product free from previously identified cloud issues (Compernolle et al., 2021, Van Geffen et al., 2022), that can be consistently combined with the most recent (beyond 14/11/2021) OFFL TrC-NO2 data."

Lines 108 - 122. Opportunity to make more concise. Current text could be shortened to 2-3 sentences. A 1-2 sentence summary of Appendix A and van Geffen et al. 2022 paper should be discussed in this paragraph. Most notably that PAL product yields larger values that the OFFL/RPRO product

We reduced the paragraph on the previous evaluations as follows : "Over the last years, the TROPOMI TrC-NO2 OFFL/RPRO products have been extensively evaluated against ground-based MAX-DOAS TrC-NO2 observations, highlighting a substantial negative bias, ranging between -23 and -37% in clean or slightly polluted areas, and increased to -51% over highly polluted areas (Verhoelst et al., 2021); regular validation updates can be found in (Lambert et al., 2022). Among other reasons (e.g. differences in representativeness, treatment of clouds and aerosols), a substantial part of this negative bias is attributed to the overly coarse (1°x1°) a priori profiles used in the S5P retrieval algorithm (Verhoelst et al., 2021), but can at least partly reduced using a priori of higher resolution, as shown for instance by Tack et al. (2021) and Douros et al. (2022). In the present study, we kept the original TROPOMI products, although we acknowledge that near future studies should probably explore such a use of alternative a priori profiles." We kept the rest of the text unchanged as we think all the elements are important. Overall, the text in this section is reduced from 51 to 38 lines.

Line 155. Mention the word "fertilizer" to be more explicit.
Done.

Line 168. No need to mention vertical level information if you are only using near-surface variables.
Done.

Line 216. Add word "long-term" before the word "values". I believe you are referring to the maximum of the May 2018 - Dec 2021 average, and not the daily maximum.
We added the information more explicitly : "On average **over the entire period (May 2018 to December 2021)**, maximum TrC-NO2 values […]"

Figure 1, 2, 3 captions - Please mention that the oversampled images are a May 2018 - Dec 2021 average.
Done.

Line 228 - Worth mentioning in this sentence the potential for this pixel to be an artefact of snow/clouds (as discussed in Appendix D)
We applied the following modification : "The historical maximum TrC-NO2 observed by TROPOMI over the Iberian Peninsula reaches the extreme value of 63 Pmolec/cm2, and occurred in early January 2021 in Madrid a few days after Filomena - the largest snowstorm since 1971 - hit central Spain (Tapiador et al., 2021), but could be at least partly impacted by some artefacts related to residual presence of snow and/or cloud shadowing effects […]". And added in Appendix D: "In addition, the residual presence of snow (unfiltered with *qa_value* threshold of 0.75) might also partly affect the extreme TrC-NO2 values observed during that period."

Line 249. Need 1-2 clarifying sentences inserted here to describe Figure 4, and why it was generated. It's not exactly clear what point you are trying to make with Figure 4. I think Lines 252 - 256 are referring to Figure 4, but this isn't clear.

We applied the following modification : "In this section, we investigate these aspects by analyzing the correlation between TROPOMI-based TrC-NO2 observations and surface NO2 mixing ratios measured by monitoring stations, **in order to highlight to which extent the variability of NO2 observed from space with TROPOMI is consistent with the one observed at the surface. Given that the aforementioned noise may impact the correlation between both variables, their correlation is here analyzed considering different averaging windows, from 1 to 365 days.**" Regarding the sentence on Fig. E5, in order to make it clearer, we move it in the first sentence of the section : "Given its relative short chemical lifetime, NO2 levels remain high close to emission sources, which typically induces a reasonable co-variability of surface NO2 mixing ratios and space-based TrC-NO2, **as illustrated for instance by both mean daily time series averaged over the entire domain (Fig. E5 in the Appendix)**."

Line 255. Remove the word "too". Also what is meant by "limited number of points"? I think you mean to say that "averaging reduces the sample size".

Yes we mean that when averaging data over a larger period, the sample size is obviously reduced. We modified as follows : "[...]and then starts to decrease, **potentially due to a more limited sample size.**"

Lines 258 - 269. Opportunity to make more concise. This could probably be shortened to 1-3 sentences.

To make it more concise, we replaced the entire paragraph by : "These results are consistent with the correlations found in the recent literature, as shown in Table 4 (note however that our results are typically based on a much larger number of points)." and reported the different correlations found in the literature in a Table.

Line 303. I see a weekly cycle in Figure 7, in that soil NOx emissions are largest on Fridays. Is this driven by a fertilizer application cycle? It does not seem that meteorological variability is the cause.

Figure 7 indeed shows a small weekly variability of soil NO emissions. This variability remains relatively small and might not be significant. Potential reasons are not clear given that we do not expect the meteorology or the fertilizer application to be responsible for such variability since no information on fertilization dates are provided to the MEGAN model.

Line 310. I also see a slight uptick on Thursday. It would be interesting to see if the TomTom data also shows upticks on Tuesdays (and Thursdays). Do you have access to any traffic data?

The TomTom data available in a few large Spanish cities (Madrid, Barcelona, Valencia) are already integrated in the construction of the anthropogenic emission profiles (shown in Fig. 7), on which no clear increase appears on these days of the week.

Figure 6. If there's room, if you can change S-NO2 to Surface-NO2 in the top label that may bring more clarity. Maybe this will mean 4 rows of text for the top label instead of 3. In the figure caption, the order of "d" and "dop" are swtiched. "dop" is listed in the third column but mentioned as the fifth in the caption.
These are good suggestions, thanks, we applied them and corrected the order error.

Line 419. What are the units person.d?
We replaced this by "person.day" in the text to make it more clear.

Section 3.5.3. What is the main takeaway point of this Section? It is not clear to me. Based on the current text, I would suggest removal of this section, but perhaps I am missing the point.
We included this section in order to further investigate why the previously analyzed monthly profiles were not showing a decrease of NO2 during the month of August, as we were expecting due to the typically lower traffic.

Figure 10. Same comment as Figure 6. If there's room, if you can change S-NO2 to SurfaceNO2 in the top label that may bring more clarity.
Done.

Lines 480 - 489. Opportunity to make more concise. These 5 sentences could probably be 2 or 3 sentences instead.
In this specific section of final discussion and conclusions, we prefer to keep the text as it is.

**Answers to referee #2**

The manuscript describes the analysis of NO2 levels over the Iberian Peninsula for different land use categories using satellite observations and ground-based data. The study focuses on the analysis of weekly and seasonal variability and their dependence on land use properties. This is a quite nice and novel approach to combine land use and air pollution information. The manuscript is suitable for publication once the authors address the following comments:

L4-5 "(considering the TrC-NO2 PAL product recently developed using a single TROPOMI 5 processor version, thus ensuring consistency over the time period)"-> Maybe this is not necessary in the abstract, you can mention the pal product but maybe avoid using 20 words on this.
We reduced the sentence as : "(considering the recently developed PAL product)".

Figure 2. Would it be possible to label the cities somehow in the figure? Not everybody is necessary familiar with the location of the cities. Also, for the administrative border: did you try white? This black on dark blue is a bit confusing. These are not a deal-breaker, just a suggestion.
Regarding the color of the administrative borders, we already dedicated some efforts to find the best compromise and we think it should remain as it is. Regarding

the names, we added a figure in the Appendix with the names of the main regions and cities, and added in the preliminary paragraph of the results section : "A map of the administrative regions and main cities over the Iberian Peninsula is provided in Fig. F1 in the Appendix."

Figure 3. Maybe you could remove the administrative borders here and leave only the city borders? Since all these administrative data are available at different resolution, you see several different lines overlapping but not precisely, and it looks a bit confusing to me.
We solved this issue with the borders in the two figures with zoom on cities.

Figure 4-5. It is not completely clear to me how do you average in time the data and why the number of data is reduced: can you clarify?
From the initial dataset at daily scale and for a given time window of X days, we first compute in each cell the sequential average over this time window, and then remove averages calculated based on (strictly) less than 50% of data. To make it clearer, let's take a simple example of a given cell and only one month of data in June [C1,C2,C3…,C30] (#pointsC = 30). When considering a time window of X=3 days, then we can compute the 3-days averages [D1,D2,D3…,D10] (#pointsD = 10).  where D1=(C1+C2+C3)/3, D2=(C4+C5+C6)/3… Then, if we assume C2 and C3 were initially missing (#pointsC = 28), then D1 will be considered as missing since it is based on less than 50% of the possible points, so finally #pointsD=9.

Also, for Fig. 4 why do you use a discrete color scale when your values are not discrete? Maybe you could change that.
In all figures, we use discrete color scales because they make it easier to identify the values (or more specifically the range of values).

3.4.1. I think it would be useful to have a map (even just in the supplement) of the land use data (or urban cover fraction) you use for the analysis to see how they relate to the distribution of NO2 over the study area. Also, it is not clear how you sampled the TROPOMI data according to land use data. How the large(r) TROPOMI pixel size is combined with the high resolution land use data? In the manuscript you write: …averaged over cells of different urban cover fractions." What you exactly mean by that? Please clarify.
We added in the Appendix two figures with the urban and crops cover fraction over the domain of study. We applied the following modification: "The mean weekly profiles of TROPOMI TrC-NO2 over the Iberian Peninsula are shown in Fig. 6, averaged over cells of different urban cover fractions **(see this urban cover fraction map in Fig. F2 in the Appendix). More specifically, TROPOMI data are averaged over 10 groups of grid cells gathered according to 10 bins of urban cover fraction (0-10%, 10-20%... 90-100%).**" Please remind that, as mentioned in the data and methods section, all datasets in this study are first regridded over a common lon-lat regular grid of 0.025°x0.025°, from which they can be easily jointly combined.

L291 "it is worth mentioning that" is redundant to me, you could maybe remove it. Overall, in the manuscript there are some quite long discussions that could be

shortened. If you manage to shorten for example some sentences in the results and conclusions, it would help in getting the main message across.

We remove this part of the sentence. Regarding the rest of the manuscript, we already applied a few modifications (suggested by other referees) that have made the text more concise.

Figure 7. The anthropogenic emissions show the largest weekend effect for the lowest bin of urban cover fraction. Can you explain that?

We applied the following modification: "Regarding the variability of emissions, the estimated total (anthropogenic and natural soil) NOx emissions indeed highlight lower emissions during the weekend, around -20%, as shown in Fig. 7; note that the strongest weekend reduction of anthropogenic emissions is found in least urbanized areas due to the predominant contribution of road transport (inter-urban roads and highways), while other emission sectors (e.g. residential) with smoother weekend reduction have a higher contribution in more urbanized areas."

**Answers to referee #3**

In their paper the authors present comparisons between gridded NO2 tropospheric column datasets derived from TROPOMI and surface observations, emission estimates and other relevant indicators like land use. Results are presented for the Iberian Peninsula. The approach and methodology is clearly not new. What is new is the degree of detail in the analysis of these comparisons and presentation of the results. I appreciated the detailed investigations of the correlations between the surface and satellite data, contrast between cities and countryside linked to different sources, weekly and seasonal variability. The figures and the text in the paper are of high quality. To my opinion this paper is an interesting information resource for environmental agencies and scientists considering to apply satellite data for local air pollution applications. As such I would be in favor of publication, after the authors have addressed my comments provided below.

We thank the referee for his/her overall positive feedback.

Comments:

line 17: "ranging from -40 % in summer to +60 % in winter ". Is this kind of variability as expected?

Yes, qualitatively, such a seasonal variability (higher NO2 in winter than in summer) is expected, notably due to both higher emissions and weaker removal by OH oxidation.

line 21: "this change started in 2018". But TROPOMI data also started in 2018, so how can a change be deduced from TROPOMI data?

This result comes from surface NO2 observations. We applied the following modification : "Some specific analysis **of surface NO2 observations** in Madrid show that the relatively sharp NO2 minimum used to occur in August".

line 33: Surface monitors can be influenced by very nearby sources and therefore they may not represent the average of a larger region. I wonder if the authors see this as a limitation?

Although this is indeed an intrinsic characteristic of in-situ point-based measurements (especially traffic or industrial point sources monitoring stations), we would not consider this as an intrinsic limitation as some studies might be interested in investigating specifically these nearby sources.

line 47: "improvement of spatial resolution of about a factor of 16" In diameter, or area?

We added the information : "about a factor of 16 **(in area)**.".

line 50: It may be useful to mention the estimate of shipping emissions, which is quite relevant for Iberia, surrounded by major shipping routes.

This sentence deals with the different applications of TROPOMI TrC-NO2 observations, so we do not see clearly the interest of providing here estimates of shipping emissions. Nonetheless, we added the detection of ship pollution plumes as an additional application of TROPOMI observations : "Among other, it has been used to map industrial point sources (e.g., Griffin et al, 2019, Beirle et al., 2021)**, detect individual ship plumes (Georgoulias et al., 2020),** […]"

line 76: I wonder if table 1 is really relevant for the paper. The topic is mapping NO2 over Iberia, not to discuss detailed pixel properties of TROPOMI. The table could be removed and summarised by providing only min-mean-max dimensions in the text.

Given that no previous studies or official documentation is providing such a detailed information on the spatial resolution of TROPOMI pixels, we do think it is useful. However, we move it to the appendix and applied the following modification: "More information on the typical dimensions of TROPOMI pixels along an orbit is given in Table A1 in the Appendix A."

line 105: "using a conservative method" Please provide more details about the gridding method.

The conservative regridding method is a common regridding approach, we simply added the reference to the *xesmf* python package used to do the regridding : "All orbit files were regridded on the target grid with the *xesmf* python package (Zhuang et al., 2022) using a conservative method […]"

line 130, GHOST: "More details on the quality assurance filtering are given in Appendix C." Again, adding this table/appendix is a bit beyond the scope of the paper to my taste. Is there a report or publication on GHOST which can be referred to and which contains this information?

Although it is still not so common in the literature, applying a careful quality control to raw observations is an important aspect of our methodology and requires to provide this information on GHOST. The reference on GHOST is currently in preparation so we are unfortunately not able to simply cite this coming paper.

line 136: The choice to look at the d1max, daily mean and overpass value (often close to the lowest values) covers the diurnal range of values, is linked to legislation and therefore I understand this choice. Comparing TROPOMI with a daily mean, however, does not seem very appropriate, given the lifetime, photochemistry and very different meteorology at night. A possible additional choice would be a window (e.g. of 6 hours) around overpass, or focusing on day-time values.

We do not claim in the paper that TROPOMI observations are representative of the daily mean NO2 but, as mentioned by the referee, this time scale was considered mainly because it is of strong interest for air pollution monitoring and regulation (as is the daily 1-hour maximum). Other time scales such as the one proposed by the referee could indeed provide other interesting insights but are of more limited interested for regulation. We clarified this choice in the text : "[…] which corresponds to the hourly NO2 mixing ratio observed around 13h30 local solar time. **Although they might not provide the best consistency with TROPOMI observations, the two first timescales are chosen for their strong interest in terms of air pollution monitoring and regulatory aspects.**"

line 191: "snow and ice at the surface". Why not? As long as snow and clouds can be reliably distinguished the retrieval should be straightforward.

The retrieval algorithm of TROPOMI does not allow to get TrC-NO2 of good quality in the presence of snow and clouds, this limitation is clearly mentioned in the TROPOMI documentation, which explains why all studies using these observations are carefully applying the recommended quality assurance filtering we also applied here (*qa_value* above 0.75).

Figure 2: Please mention the averaging period in the caption.

Done.

line 233 etc.: These summer-winter sampling differences and impact on the yearly mean may be (partly) overcome by computing monthly-mean values, and then averaging over the months. As long as the sample is big enough such seasonally-varying sampling numbers should not be a real problem.

The referee is right that this could partly limit the differences of sampling between summer and winter.

Here, the bias we are mentioning is related to the fact that no NO2 observations are available under specific meteorological conditions (more specifically, cloudy conditions) that might impact the sources and sinks of NO2 in a different way compared to clear-sky conditions. For instance, regarding the anthropogenic emissions, cloudy conditions could reduce the production of solar energy, which might induce stronger emissions from natural gas power plants, but could also be associated with stronger winds, thus favoring the production of electricity from wind turbines. Also, when such cloudy conditions come with rainfall, a part of the population might favor the use of their car against soft transportation modes (e.g., walk, bike). Natural soil NO emissions also depends on the meteorology. Regarding the sinks, cloud conditions are expected to reduce the oxidation of NO2 by OH, which can increase the NO2 chemical lifetime and thus the NO2 mixing ratios. On the other side, some studies reported non-negligible scavenging of NO2 during rain

events, which could reduce the NO2 concentrations. Therefore, many factors could play a role in this sampling-based bias. Note that even when computing the annual mean from the monthly mean, results (including the bias) are unchanged.

Fig. 5, caption: It would be helpful for the reader to repeat what d, dop, d1max mean.
Done.

line 246: Is it really TROPOMI noise, or actual spatio-temporal variability in NO2? Is it possible to distinguish these two? Same question for lines 254-255.
Although it is greatly reduced compared to previous space sensors, TROPOMI TrC-NO2 observations remain affected to some extent by noise. To the best of our knowledge, quantifying which part of this apparent noise is related to actual spatio-temporal variations of NO2 is not possible, the only available option consisting in averaging the observations over a larger time window in order to increase the signal-to-noise ratio. Concerning the slight decrease of correlation beyond time windows of 90-120 days, we indicated that this could be due to the lower number of points because we could not find a good reasons for this correlation to decrease, but other factors might play a role here.

line 254: i.e. 3-4 months
Corrected.

line 257: d1max seems to have a much lower PCC!
Yes indeed, this is already mentioned in the sentence.

Fig. E4: It is difficult to understand what is plotted here.
We applied the following modification: "Mean TrC-NO2 coincident with the 10 \% largest residuals of the daily-scale TrC-NO2 versus surface NO2 linear regression, **these high residuals corresponding to situations where strong TrC-NO2 were measured by TROPOMI, while comparatively lower NO2 mixing ratios were measured by surface background stations (see text in Sect. 3.3).**"

line 357: "Interestingly, this mean TrC-NO2 weekend effect progressively decreases when focusing on largest industrial point sources, " I was wondering if this is really an effect of the large industries continued activity over the weekend, or if this is caused by a relatively larger contribution to the column in comparison to the inflow from residential areas.
Most of the industries are expected to reduce their activity and thus their NOx emissions during the weekend, although this reduction likely varies from one type of industry to another. Some industries might completely turn off their activity while others (like power plants) are only slowing down in order to follow the lower electricity demand during the weekend. Detailed information on these aspects are lacking, and current temporal profiles used to temporally disaggregate annual emissions remain affected by substantial uncertainties. As underlined in the text, these TROPOMI-based weekly profiles above industrial areas are likely at least partly influenced by the short/mid-range transport of NO2 plumes from urban areas.

Therefore, without more detailed analysis, it remains difficult to provide more robust interpretation of the weekly effect obtained over these industries.

line 377: "slightly stronger discrepancies over least urbanized areas". Reasons for this could be the relatively larger importance of NO2 higher in the atmosphere, NO2 from other sources and importance of inflow from elsewhere.
Indeed all the factors may play a role but we prefer here to avoid commenting too much as in absolute terms, the variability among weekdays remains quite weak in these least urbanized areas.

line 391: "results highlight an increase of the weekend effect over the last two decades" This is surprising. Please try to explain: could it be linked to social factors (people working less in the weekend)?
We added the following paragraph: "Reasons for such a trend are not clearly identified but according to the official national inventory of Spain, from 2000 to 2020, the relative contribution of the road transport sector remained unchanged (35%), slightly increased in the industry sector (from 14 to 16%) and in the other agricultural sources (from 7 to 12%), and strongly decreased in the public power sector (from 21 to 5%). This last decrease resulted from several European Union regulations (e.g., Directive 2001/80/EC, Directive 2010/75/EU) and the shift from coal to natural gas in the electricity sector. In terms of emission weekly profile, the weekend reduction in the public power sector is much weaker than in other major emission sectors such as road transport (-25% on Saturday-Sunday compared to the mean Monday-Friday, against -57% for road transport). Therefore, its decreasing relative contribution may explain at least part of the increasing weekend effect of NO2 concentrations observed at the surface. In addition, heavy-duty vehicles have been ban from circulating in certain highways and entering main Spanish urban areas on Sundays (e.g., Real Decreto 1428/2003), which may have also contributed to the increasing weekend effect, at least on this specific day."

line 411: "discrepancies" This word has a negative meaning. Maybe "difference" is better to use. There may be several good reason why the detailed seasonal dependence may differ at the surface compared to the column even if the observation is perfect.
We agree this term has a negative meaning, we replaced it in the entire manuscript (except in one occurrence where it refers to the evaluation of CAMS against TROPOMI).

line 415: "such a broad flat minimum over most urbanized areas was not expected given that road transport in many cities is known to be substantially reduced in August when many people go on holidays," When I look at the curve it seems that the biggest difference occurs in April-May where the relative satellite column is higher than surface observations. June-August looks quite consistent at the surface and from space. A more focused discussion for April-May would be interesting.
Here we are not discussing the differences between TROPOMI TrC-NO2 and surface NO2 (which are indeed higher in April-May than in June-August), but the broad flat minimum itself (compared to the drop in August we were initially expecting). In any

case, following the suggestion of another referee, we move this section in the Appendix and only briefly mention its main result in the previous section.

line 432-435: I would challenge the authors to find a more detailed explanation for the shift from August to May. A shift in chemistry (linked to the downward trend in NO2) or anomalous meteorology are factors to look at. Could surface ozone provide additional clues?

Looking at the surface O3 observations and ERA5 reanalysis data, we could not find anomalous conditions able to explain this result. We modified this paragraph as follows : "Reasons for this relative flattening of the seasonal minimum of NO2 remain unclear but may include the continuously decreasing contribution of road transport to total NOx emissions in Madrid, and the associated increasing relative contribution of other emission sectors presenting a distinct monthly variability (and no clear drop in August). Specifically on May 2019, the relatively low NO2 mixing ratios observed at the surface do not appear to be explained by specific meteorological conditions (monthly anomalies of surface temperature and geopotential at 500 hPa in the ERA5 reanalysis are only +0.8° and +150 dam compared to the 2012-2021 climatology). Some specific chemical conditions might eventually play a role, but mean surface O3 mixing ratio during this month remained very close (+0.5 ppbv) to their climatological value.". In any case, note that all this discussion has been moved to the Appendix.

Table 4: TROPOMI column data between 2018 and 2021 could be added here.

We added the TROPOMI monthly mean above Madrid in the Table and the following text: "Note that TROPOMI observations above Madrid (Table F1, bottom part) show a quite similar picture as the one obtained in areas exceeding 90% of urban cover fraction (Sect. 3.5). Excluding the year 2020, a broad minimum is typically observed in summer. The minimum of 2019 found here in November is likely not representative due to an especially high cloud cover during this specific period (explaining the small number of days with available observations). In 2021, the minimum occurs in August but with TrC-NO2 values close to those observed in June and July.".

line 449 etc: Is coverage really an issue to worry about? I do not find a clear answer to this in the paper. Even with 30% coverage there are 10 observations in a month at a given location. The variability in concentrations (linked e.g. to meteorological variability) and sampling should be compared with the amplitude of signals to be picked up. In combination with spatial or temporal averaging, as presented in the figures in the paper, it seems that sampling+observation noise is not a problem, and spatial/weekly/seasonal/trend estimates seem all very reliable.

As explained in a previous answer, to our opinion, as soon as data gaps are not randomly distributed in time (as it is the case for TROPOMI), they can potentially introduce some bias when calculating (monthly or annual) averages.

line 464: "lies in its observations over rural areas and seas." I found the analysis of shipping signals underrepresented / missing in the paper. A separate section on this (figure) would be good.

Although we fully agree that it is an interesting and important topic on which TROPOMI can provide observations of strong interest (as already underlined in the paper), such analysis would require a substantial amount of additional work, notably to objectively distinguish the signal coming from shipping emissions and the one advected from coastal land areas. A comprehensive analysis of these aspects would ideally require information on the shipping traffic. Also, the present paper is strongly focusing on the analysis of the consistency of TROPOMI observations with in-situ observations available at the surface, which would not be possible over the sea. Therefore, we do think that this analysis falls beyond the scope of our study.

line 496: "discrepancies" Again, I'm not sure if this is the correct word. Surface concentrations are influenced by different factors than column amounts.
As previously mentioned, we replaced this term.

line 502: It would be useful to add some extra references to methods estimating surface concentrations (e.g. Cooper, https://doi.org/10.1088/1748-9326/aba3a5) and emissions from satellite column observations. These approaches + references are not mentioned sufficiently to my opinion.
Our study aims at providing a comprehensive analysis of the TROPOMI TrC-NO2 dataset and its co-variability with surface NO2 concentrations. To our opinion, although interesting by themselves, the studies like the one of Cooper et al. (2020) are not of direct interest here as they focus more on exploiting rather than analyzing this co-variability, in order to get the most accurate estimates of surface NO2. Nonetheless, we agree it is worth mentioning them in the introduction: "[…] analyse lightning NOx emissions (Perez-Invernon et al., 2022). **In addition, TROPOMI TrC-NO2 observations have been used in several studies to infer surface NO2 concentrations, using geo-statistical (e.g., Zhang et al., 2022) or geophysical (e.g., Cooper et al., 2020) models.**"